# SCALING ATTENTION VIA FEATURE SPARSITY

**Yan Xie**[1*]   **Tiansheng Wen**[1,2*]   **Tangda Huang**[1]   **Bo Chen**[1†]   **Chenyu You**[2]
**Stefanie Jegelka**[3,4]   **Yifei Wang**[5†]
[1] School of Electronic Engineering, Xidian University   [2] Stony Brook University
[3] TUM   [4] MIT   [5] Amazon AGI SF Lab[‡]

## ABSTRACT

Scaling Transformers to ultra-long contexts is bottlenecked by the $O(n^2 d)$ cost of self-attention. Existing methods reduce this cost along the sequence axis through local windows, kernel approximations, or token-level sparsity, but these approaches consistently degrade accuracy. In this paper, we instead explore an orthogonal axis: *feature sparsity*. We propose **Sparse Feature Attention (SFA)**, where queries and keys are represented as $k$-sparse codes that preserve high-dimensional expressivity while reducing the cost of attention from $\Theta(n^2 d)$ to $\Theta(n^2 k^2/d)$. To make this efficient at scale, we introduce **FlashSFA**, an IO-aware kernel that extends FlashAttention to operate directly on sparse overlaps without materializing dense score matrices. Across GPT-2 and Qwen3 pre-training, SFA matches dense baselines while improving speed by up to $2.5\times$ and reducing FLOPs and KV-cache by nearly 50%. On synthetic and downstream benchmarks, SFA preserves retrieval accuracy and robustness at long contexts, outperforming short-embedding baselines that collapse feature diversity. These results establish feature-level sparsity as a complementary and underexplored axis for efficient attention, enabling Transformers to scale to orders-of-magnitude longer contexts with minimal quality loss. Code is available at https://github.com/YannX1e/Sparse-Feature-Attention.

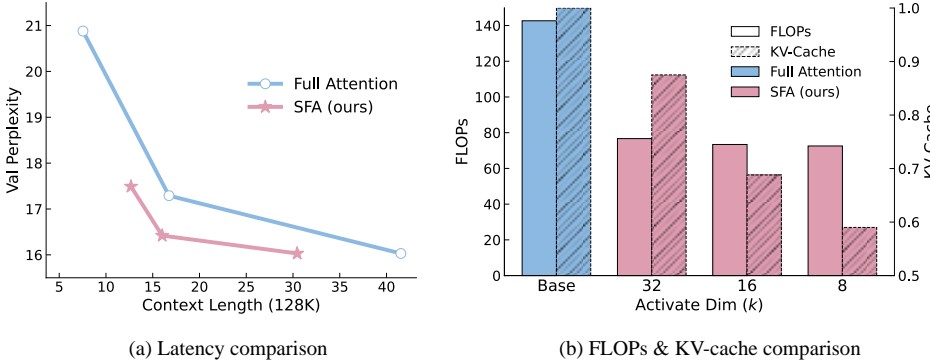

(a) Latency comparison                    (b) FLOPs & KV-cache comparison

Figure 1: **Overview of our proposed method.** (a) Trade-off between performance and speed. Compared to directly reducing dimensionality with *short embeddings*, our method achieves a more favorable balance, delivering a 259% speedup over the original dimensionality while improving performance by 21.4% relative to the short-embedding baseline. (b) Computational and memory efficiency comparison. Our method reduces KV-cache memory usage by 41% and FLOPs by 49%.

[*]Equal Contribution.

[†]Corresponding Authors: Bo Chen (bchen@mail.xidian.edu.cn) and Yifei Wang (yifeiwg@amazon.com).

[‡]This work was done at MIT prior to Yifei Wang joining Amazon.

# 1 INTRODUCTION

Scaling language models to ever longer contexts is fundamentally limited by the $O(n^2 d)$ cost of self-attention, where $n$ is the sequence length and $d$ the feature dimension. Most existing approaches attempt to reduce this cost along the sequence axis. Windowed or low-rank attention variants constrain interactions to achieve linear complexity, while token-level sparsity prunes which tokens interact (Child et al., 2019; Beltagy et al., 2020; Zaheer et al., 2020; Choromanski et al., 2021; Wang et al., 2020; Xiong et al., 2021). Yet large-scale benchmarks consistently show that these approximations sacrifice accuracy, leaving dense attention the most reliable option at long ranges. This raises a natural question: *rather than reducing the set of tokens, can we explore feature diversity as an orthogonal axis for scaling attention?*

This question is motivated by findings in representation learning, where sparse embeddings (Formal et al., 2021; Wen et al., 2025; Guo et al., 2026; You et al., 2025; Duan et al., 2024; Xie et al., 2025) show that high-dimensional spaces encode rich features and that selective activation can preserve expressivity while yielding large efficiency gains. If attention itself can be viewed as retrieval over feature coordinates, then sparsifying queries and keys by activating only their most salient dimensions could reduce computation without collapsing representational capacity. The challenge is to realize this idea in practice: how to preserve expressivity while sparsifying, how to implement kernels that benefit from sparsity without materializing the $n \times n$ score matrix, and how to adapt pretrained dense models without eroding their quality.

We address these challenges with **Sparse Feature Attention (SFA)**. Instead of dense $d$-dimensional queries and keys, SFA learns $k$-sparse codes in which each token activates only a handful of coordinates. Attention scores are computed solely from overlaps between these supports, reducing the arithmetic of $QK^\top$ from $\Theta(n^2 d)$ to $\Theta(n^2 k^2/d)$ – a fraction $(k/d)^2$ of the dense cost – while storing only $O(nk)$ nonzeros. To make this efficient at scale, we introduce **FlashSFA**, a new IO-aware kernel that extends FlashAttention by operating directly on sparse overlaps with online softmax. This design avoids materializing any dense $n \times n$ scores, retains exactness, and brings compute and memory scaling in line with feature sparsity.

The benefits of this shift are demonstrated in Figure 1. Compared to simply shrinking hidden size ("short embeddings"), SFA achieves a much better trade-off: it improves perplexity by more than 20% while delivering over $2.5\times$ speedup, and reduces FLOPs by nearly half together with a 41% drop in KV-cache memory. Experiments confirm that these benefits extend broadly. On GPT-2 and Qwen3 pretraining, SFA matches dense baselines in perplexity and downstream accuracy. On synthetic long-context benchmarks such as Needle-in-a-Haystack, it sustains retrieval accuracy across unseen lengths, while providing consistent latency gains. Crucially, the method is orthogonal to token-level sparsity and paging, multiplying their benefits by lowering per-interaction cost.

This work thus establishes feature-level sparsity as a powerful and previously underexplored axis for efficient attention. By leveraging feature diversity rather than compressing it away, SFA preserves high-dimensional expressivity while unlocking substantial efficiency gains. Together with FlashSFA, it makes exact long-context attention practical at scale, and paves the way for context windows extended by orders of magnitude without compromising model quality.

# 2 PRELIMINARIES

**Transformers and multi-head attention.** Let a sequence of $n$ tokens be represented by hidden states $X \in \mathbb{R}^{n \times d_{\text{model}}}$. For each head $h \in \{1, \dots, H\}$ with head dimension $d$, standard scaled dot-product attention computes:

$$Q_h = XW_h^Q \in \mathbb{R}^{n \times d}, \qquad K_h = XW_h^K \in \mathbb{R}^{n \times d}, \qquad V_h = XW_h^V \in \mathbb{R}^{n \times d}, \quad (1)$$

$$S_h = \frac{Q_h K_h^\top}{\sqrt{d}} \in \mathbb{R}^{n \times n}, \qquad P_h = \text{softmax}(S_h \odot M) \in \mathbb{R}^{n \times n}, \qquad O_h = P_h V_h \in \mathbb{R}^{n \times d}, \quad (2)$$

where $M$ encodes causal or padding masks, and the head outputs are concatenated and projected. The principal cost arises from the dense $Q_h K_h^\top$ and materialization of $P_h$; IO-aware kernels (e.g., FlashAttention) compute $O_h$ in tiles without forming $P_h$ explicitly, minimizing HBM traffic while remaining exact (Dao et al., 2022; Dao, 2024; Shah et al., 2024).

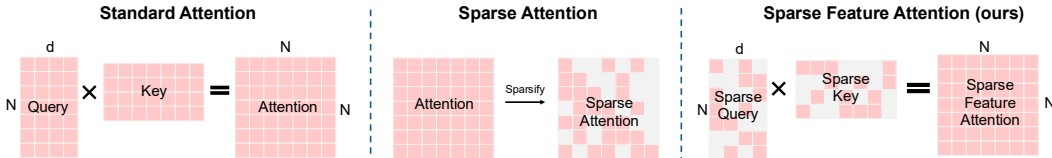

Figure 2: **Three paradigms of attention.** *Left: Standard attention* computes all $N \times N$ query–key interactions in the full feature dimension $d$. *Middle: Sparse attention* reduces cost by selecting, for each query $i$, a small subset of keys $\Omega_i$ and masking the remaining logits before softmax, but each retained interaction still spans all $d$ features. *Right: Sparse Feature Attention (ours)* keeps all tokens but sparsifies along the feature axis by selecting the top-$k$ channels in $Q$ and $K$ ($\tilde{Q} = \text{Topk}_k(Q), \tilde{K} = \text{Topk}_k(K)$). Attention is then computed only over overlapping selected features with sparse matrix multiplication. This shifts sparsity from the token axis ($N \times N$) to the feature axis, achieving efficiency while preserving token coverage.

**Sparse formats for efficient storage.** Sparse matrices that contain only a few non-zero elements can be stored efficiently in sparse formats. Consider a matrix $A \in \mathbb{R}^{n \times d}$ with $\text{nnz}(A)$ nonzero elements. In the *Compressed Sparse Row* (CSR) format, we store three arrays: (i) data $\in \mathbb{R}^{\text{nnz}(A)}$, containing the values of all nonzero entries, (ii) indices $\in \{0, \dots, d-1\}^{\text{nnz}(A)}$, recording the column index of each nonzero, and (iii) indptr $\in \{0, \dots, \text{nnz}(A)\}^{n+1}$, where indptr[i] marks the offset in data/indices where row $i$ begins. Thus, the nonzeros of row $i$ can be read quickly from data[indptr[i]:indptr[i+1]]. The *Compressed Sparse Column* (CSC) format is analogous, but compresses by columns instead of rows, with an indptr array of length $d + 1$ (Saad, 2003; Davis, 2006).

**Efficient multiplication with spare formats.** When multiplying two sparse matrices, the cost is not proportional to the dense size $n \times d$ but rather to the number of *structural intersections* between the nonzero patterns of rows and columns. This operation, called Sparse General Matrix Multiplication (SpGEMM), is typically implemented by Gustavson's row-wise accumulation algorithm (Gustavson, 1978) or by hash-based methods (Buluc & Gilbert, 2011). The efficiency of SpGEMM therefore depends on how many row–column index sets overlap, making CSR and CSC natural formats for storing query and key matrices in our method.

## 3 Sparse Feature Attention

This section introduces *Sparse Feature Attention* (SFA), a drop-in modification of multi-head self-attention that operates along the *feature* axis. Each query/key vector is converted into a $k$-sparse code; attention scores are then computed *only* on overlapping active coordinates. This preserves the probabilistic semantics of exact softmax attention over learned supports while reducing arithmetic, memory traffic, and KV-cache growth.

### 3.1 Attention via Sparse Matrix Multiplication

The key idea of SFA is to sparsify the query and key features before attention computation, so that only their most salient coordinates contribute to similarity scores. As illustrated in Figure 2 (right), given dense projections $Q, K, V \in \mathbb{R}^{n \times d}$, we apply a row-wise Top-$k$ operator to both $Q$ and $K$:

$$\tilde{Q} = \text{Topk}_k(Q), \qquad \tilde{K} = \text{Topk}_k(K), \tag{3}$$

where for $x \in \mathbb{R}^d$,

$$\text{Topk}_k(x)_u = \begin{cases} x_u, & u \in \arg \text{topk}(|x|), \\ 0, & \text{otherwise.} \end{cases} \tag{4}$$

Thus each query and key vector is converted into a $k$-sparse representation, preserving only its $k$ largest-magnitude entries. These $\tilde{Q}, \tilde{K}$ serve as sparse query and key features for attention.

**Sparse attention via sparse matrix multiplication.** Attention scores are then computed as $S = \tilde{Q}\tilde{K}^\top$. Instead of full dense multiplication, we exploit sparsity: each nonzero in $\tilde{q}_i$ interacts only

with keys that share the same active coordinate. For query $i$ with support $S_i$,

$$s_{ij} = \frac{1}{\sqrt{d}} \sum_{u \in S_i \cap S_j} \tilde{q}_{i,u}\, \tilde{k}_{j,u}, \tag{5}$$

which corresponds to sparse matrix multiplication between $\tilde{Q}$ (CSR format) and $\tilde{K}^\top$ (CSC format). Traversing active coordinates yields only the nonzero attention edges. The resulting scores are then passed through the usual softmax and value aggregation steps.

**Backward computation.** Leveraging the sparse structure, we can also skip computing the gradient for the full query and key matrices at backward computation. Specifically, we use a straight-through estimator: gradients flow back only through the selected coordinates. For query $i$ with support $S_i$,

$$\frac{\partial \mathcal{L}}{\partial q_{i,u}} = \begin{cases} \frac{\partial \mathcal{L}}{\partial \tilde{q}_{i,u}}, & u \in S_i, \\ 0, & u \notin S_i, \end{cases} \tag{6}$$

and similarly for $k_{j,u}$. Both forward and backward passes scale only with the sparse edge set.

**Efficiency analysis.** Dense attention requires $\Theta(n^2 d)$ computation and $\Theta(n^2)$ memory, since every query interacts with every key across all $d$ feature dimensions. In contrast, SFA only forms scores along feature coordinates selected by both queries and keys. Each token activates $k$ features, giving $nk$ nonzeros in total. Assuming supports are balanced across dimensions, each coordinate is chosen by about $\deg(u) \approx nk/d$ tokens. The number of query–key overlaps contributed by coordinate $u$ is then $\deg(u)^2$, and summing over all $d$ coordinates yields:

$$E \approx \sum_{u=1}^{d} \deg(u)^2 \approx d\left(\frac{nk}{d}\right)^2 = \frac{n^2 k^2}{d}. \tag{7}$$

Thus the total cost for attention shrinks from $\Theta(n^2 d)$ (dense) to $\Theta(n^2 k^2/d)$ (sparse), which is only a fraction $k^2/d^2$ of the dense cost. Both forward and backward passes then cost $O(E + Ed_v)$ FLOPs, and memory for storing query and key drops from $O(nd)$ to $O(nk)$ with the sparse formats. For concreteness, with $d = 128$ and $k = 16$ (default setting considered in this work), the ratio is $k^2/d^2 = 1/64$, i.e. about a $64\times$ reduction in theory. As the dimension $d$ increases in larger models, the gain could be even higher. For $d = 1024$ and $k = 32$ (shown to have very similar retrieval performance in Wen et al. (2025)), the ratio is $32^2/1024^2 = 1/1024$, i.e., a reduction of more than $1000\times$. This means sparse feature attention can potentially extend context length by one to three orders of magnitude at similar compute cost. For example, turning a 1M context window into 64M or even 1G, opening up substantial improvements for long-context applications.

## 3.2 FlashSFA: Fast Sparse Feature Attention without Materialization

A key challenge in Sparse Feature Attention (SFA) is that, although we reduce the number of pairwise interactions from $n^2 d$ to $n^2 k^2/d$, a naïve implementation would still require materializing an $n \times n$ score matrix to apply the softmax. This would destroy the memory advantage, as the $O(n^2)$ storage is often the real bottleneck at long sequence lengths.

FlashAttention addressed exactly this issue in the dense case: it avoids storing $QK^\top$ by processing queries and keys in small tiles, keeping only a temporary tile buffer of partial scores on-chip. An online softmax update maintains numerical stability and exactness without ever writing the full $n \times n$ matrix to memory (Dao et al., 2022). FlashAttention-2 and -3 extend this idea with more parallelism and precision refinements (Dao, 2024; Shah et al., 2024). Our proposed **FlashSFA** extends this principle to SFA. We retain the IO-aware tiling and online-softmax machinery of FlashAttention, but replace dense tile multiplications with sparse feature-intersection kernels. For a tile of queries (rows $i \in [i_0, i_0 + B_r)$) and keys (columns $j \in [j_0, j_0 + B_c)$), the kernel iterates over the active features of these tokens, intersects their supports, and performs scatter-adds into a compact $B_r \times B_c$ score buffer. This buffer is immediately consumed by the online softmax update, so no large score matrix is ever written to memory. The result is mathematically identical to computing $\mathrm{softmax}(\tilde{Q}\tilde{K}^\top/\sqrt{d})V$, but with both compute and memory scaling as in SFA.

**Efficiency and design.** FlashSFA inherits the same $O(n)$ IO complexity of FlashAttention, since only tiles (not the full matrix) touch high-bandwidth memory. Within each tile, the work is proportional to the number of overlapping features rather than $d$, yielding the $O(n^2 k^2/d)$ complexity

Table 1: **Perplexity and Accuracy results.** Dense baselines use full hidden size and uncompressed KV cache; "Dense ($d$=X)" denotes short-embedding baselines with reduced feature dimension. PPL is evaluated on **OpenWebText** for GPT-2 and **Pile** for Qwen3. Note that "Dense (full)" serves as a reference upper bound; we highlights the best results among the sparse/compressed baselines.

| Model | | Latency ↓ | PPL ↓ | Acc ↑ | | | | | |
|---|---|---|---|---|---|---|---|---|---|
| | | 128k context | OWT/Pile | PiQA | LAMBADA | ARC-e | ARC-c | HellaS | Avg-Acc |
| GPT2-124M | Dense (full) | 16.86 | 17.29 | 56.34 | 22.78 | 28.35 | 14.32 | 19.61 | 28.28 |
| | Dense ($d = 32$) | 7.86 | 20.88 | 51.30 | 19.39 | 25.72 | 12.47 | 14.26 | 24.63 |
| | SFA ($k = 8$) | 9.41 | **18.27** | **54.92** | **21.03** | **28.41** | 7.39 | **19.26** | **27.40** |
| GPT2-350M | Dense (full) | 46.78 | 15.03 | 59.79 | 24.74 | 30.19 | 15.78 | 22.04 | 30.51 |
| | Dense ($d = 32$) | 20.58 | 19.89 | 55.17 | 19.96 | 28.15 | 11.83 | 18.43 | 26.71 |
| | SFA ($k = 8$) | 23.67 | **16.78** | **58.02** | **23.83** | **30.22** | **13.66** | **22.13** | **29.57** |
| Qwen3-0.6B | Dense (full) | 77.65 | 4.66 | 62.47 | 34.82 | 45.41 | 20.35 | 33.95 | 39.40 |
| | Dense ($d = 64$) | 30.84 | 6.03 | 58.43 | 31.27 | 41.58 | 15.83 | 28.29 | 36.68 |
| | SFA ($k = 16$) | 34.20 | **4.81** | **61.73** | **34.05** | **45.62** | **19.27** | **34.03** | **38.94** |

analyzed in §3.1. The online softmax logic, masking for causality, and streaming of $V$ are unchanged. Indices for sparse features add modest overhead ($O(nk)$), and can be stored efficiently with 16-bit integers for typical $d \leq 65{,}535$.

By marrying the sparsity of SFA with the memory-efficient tiling of FlashAttention, FlashSFA achieves the best of both worlds: it avoids $O(n^2)$ materialization while preserving the $\frac{k^2}{d^2}$ reduction in arithmetic and memory cost. This enables exact attention with dramatically lower compute and memory footprints, making long-context training and inference practical at scale. We defer a full description of the FlashSFA algorithm to Appendix C.

## 4 EXPERIMENTS

### 4.1 PRETRAINING EXPERIMENTS

Having introduced Sparse Feature Attention (SFA) and the FlashSFA kernel, we next examine whether autoregressive LMs trained from scratch can maintain modeling quality under feature sparsification. We evaluate GPT-2 and Qwen3 models against dense and short-embedding baselines, measuring both modeling quality and efficiency.

**Models and baselines.** We study GPT-2 Small/Medium (Radford et al., 2019) and Qwen3-0.6B (Yang et al., 2025), replacing dense $QK^\top$ scoring with SFA while keeping $V$ dense. Sparsity budgets $k \in \{8, 16\}$ are tested. Baselines include standard dense attention and short-embedding variants (halving the hidden size of $Q/K$). Note that "Dense (full)" serves as a reference upper bound; we highlights the best results among the sparse/compressed baselines. We use the RTopK kernel (Xie et al., 2024) for efficient topk operations. Additional implementation details, including model configurations and handling of RoPE dimensions in Qwen3, are deferred to Appendix A.1.

**Datasets and benchmarks.** GPT-2 models are trained on OpenWebText (Gokaslan & Cohen, 2019), Qwen3 on The Pile (Gao et al., 2020; Biderman et al., 2022). We report validation perplexity (PPL), zero-shot accuracy on PiQA (Bisk et al., 2020), LAMBADA (Paperno et al., 2016), ARC-e/ARC-c (Clark et al., 2018), and HellaSwag (Zellers et al., 2019), as well as decoding throughput at 128k tokens (`Speed@128k`) to assess long-context efficiency.

**GPT-2 results.** Table 1 shows that SFA with $k = 16$ (not shown here but consistent with $k = 8$ trends) closely tracks dense baselines, with negligible differences in perplexity and accuracy. SFA with $k = 8$ incurs slightly higher PPL and minor accuracy drops, but these remain within acceptable bounds. This demonstrates that sparsified features preserve most of the model's expressive capacity. By contrast, short-embedding baselines degrade more substantially: they reduce perplexity efficiency and underperform on challenging tasks such as ARC-c, especially for GPT-2 Small. While such baselines deliver higher throughput, their quality–efficiency balance is skewed toward

Table 2: **Long context pretraining results.** Comparison of NIAH accuracy rates for different lengths under various training lengths. (a) Models are trained on 8k synthetic NIAH data, and the accuracy rate on test lengths from 1k to 8k. (b) Models are trained on 32k synthetic NIAH data, and the accuracy rate on test lengths from 1k to 32k.

(a) NIAH accuracy (%) within 8k Sequence Length.

| Method | Context Length | | | | Speed@8k |
|---|---|---|---|---|---|
| | 1k | 2k | 4k | 8k | |
| Dense ($d = 64$) | 94% | 93% | 90% | 95% | 1.0× |
| SFA ($k = 2$) | 95% | 95% | 97% | 98% | **1.9×** |
| SFA ($k = 8$) | **98%** | **100%** | **99%** | **98%** | 1.3× |

(b) NIAH accuracy (%) within 32k Sequence Length.

| Method | Context Length | | | | Speed@32k |
|---|---|---|---|---|---|
| | 1k | 4k | 16k | 32k | |
| Dense ($d = 64$) | 92% | 94% | 83% | 80% | 1.0× |
| SFA ($k = 8$) | 95% | 94% | 83% | 82% | **1.3×** |
| SFA ($k = 16$) | **97%** | **96%** | **83%** | **83%** | 1.0× |

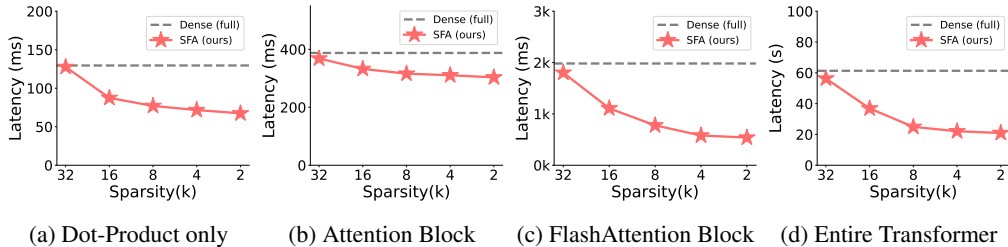

(a) Dot-Product only  (b) Attention Block  (c) FlashAttention Block  (d) Entire Transformer

Figure 3: **Latency vs. feature sparsity.** Latency Comparison of dense attention and SFA (ours) at different modular levels in Transformers under 16k context length. Higher sparsity brings substantial decrease in latency.

speed, making them less appealing. On retrieval-like tasks (LAMBADA, HellaSwag), sparse models underperform relative to their PPL, motivating further retrieval-focused experiments (Section 4.2).

**Qwen3 results.** For Qwen3-0.6B, also in Table 1, SFA with $k = 8$ maintains perplexity nearly identical to dense (4.81 vs. 4.66) and preserves accuracy across PiQA, ARC-e, and HellaSwag. The small differences on ARC-c (19.27 vs. 20.35) and average accuracy (38.94 vs. 39.40) suggest only a marginal quality cost. Short-embedding baselines again degrade more severely, with higher PPL (6.03) and lower accuracy (Avg-Acc 36.68). This confirms that even in modern architectures with RoPE and normalization refinements, sparsified features remain competitive with dense attention, while offering clear efficiency benefits at long context.

**Efficiency results.** Across GPT-2 and Qwen3, short-embedding variants provide the largest raw speedups due to narrower hidden size, but their accuracy loss makes them less practical. Sparse models present a more balanced trade-off: $k = 16$ maintains baseline-level quality, and $k = 8$ provides moderate speedups while remaining close in accuracy. In practice, $k = 8$ emerges as the most attractive setting, balancing efficiency and modeling quality. This setting is therefore used in subsequent scaling and efficiency benchmarks (Section 4.3).

## 4.2 SYNTHETIC NIAH EXPERIMENTS

The synthetic *Needle-in-a-Haystack* (NIAH) benchmark provides a controlled way to examine how models handle extremely long contexts and retrieval-style reasoning. To further examine whether sparse attention preserves retrieval capacity over long contexts, we conduct experiments on the synthetic NIAH task. Following the RULER methodology, haystacks are constructed by repeating the character "#" and inserting a single target "needle" token that the model must recover. We train GPT-2 models (124M) from scratch on synthetic NIAH QA data under two training regimes: one restricted to 8k contexts and one extended to 32k contexts. In both cases, we then evaluate test accuracy across multiple held-out lengths, measuring how well models generalize beyond their training window. Speed is also measured at the maximum training length to capture efficiency.

**Results within 8k.** Table 2a reports results when models are trained up to 8k tokens. Dense baselines perform well across all lengths but incur standard compute costs. Sparse models not only match but slightly exceed dense accuracy, achieving near-perfect recovery at all test lengths. At the

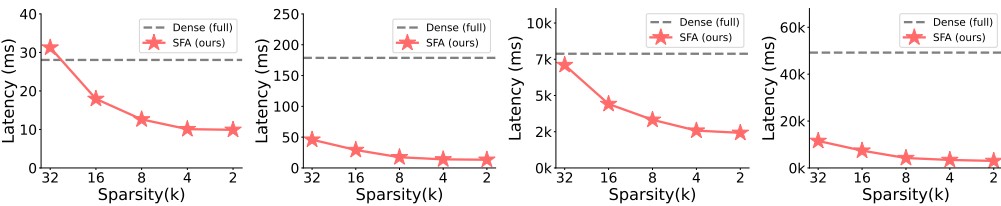

(a) 128 dim (4k context)   (b) 256 dim (4k context)   (c) 128 dim (65k context)   (d) 256 dim (65k context)

Figure 4: **Latency vs. feature sparsity with various config.** Latency Comparison of dense attention and SFA (ours) at different head dimensions and context lengths. Notably, the latency of SFA can be much lower than dense attention under high dimension per head and long context, e.g., Figure 4d.

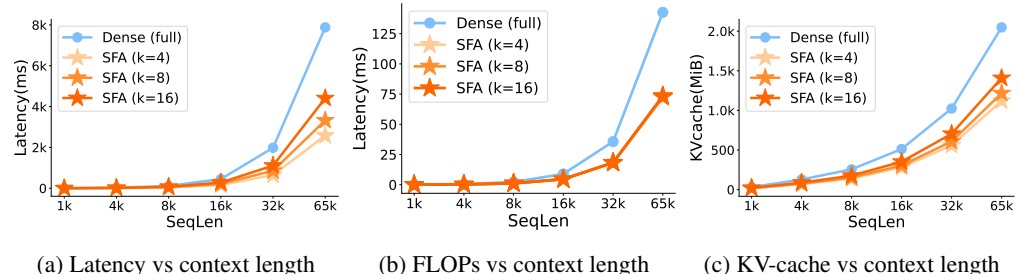

(a) Latency vs context length   (b) FLOPs vs context length   (c) KV-cache vs context length

Figure 5: **Scaling dense attention and SFA with context length.** SFA can consistently reduce both the computatin cost and KV cache size by a constant factor of at least 2.

same time, SFA delivers a **1.9**× decoding speedup at 8k for $k = 2$, confirming that sparse scoring reduces computation without sacrificing reliability.

**Results within 32k.** Table 2b extends training to 32k tokens. Dense baselines degrade as length grows, dropping to 80% accuracy at 32k. SFA models maintain higher accuracy: $k = 8$ holds steady at 82% and $k = 16$ at 83%. Notably, $k = 8$ delivers **1.3**× faster generation at 32k, while $k = 16$ matches dense throughput. These results show that sparse attention generalizes robustly across unseen lengths while simultaneously reducing long-context latency.

**Discussion.** The NIAH task isolates retrieval in a controlled setting, making it possible to compare dense and sparse features without confounding factors. Across both 8k and 32k training regimes, SFA preserves or improves accuracy while achieving consistent speedups. This complements the pretraining results in Section 4.1: sparse attention does not erode retrieval ability, and under synthetic stress tests it can even provide stronger length generalization than dense attention.

### 4.3 BENCHMARKING COMPUTATION AND MEMORY EFFICIENCY OF SFA

We benchmark Sparse Feature Attention (SFA) in both training and inference scenarios, since they stress different system bottlenecks. Training-time attention is dominated by quadratic computation, while inference-time attention with KV cache is dominated by memory traffic. Experiments are run on an *A800* GPU with *CUDA 12.4*, using INT32 for `indptr` and INT8 for `indices`, FP16 for `values`, and pinned batches in HBM. Timing excludes dataloader overhead. All kernels are compiled with *CUDA and Ninja*, and we report medians over 50 warm runs. We built our FlashSFA kernel upon LeetCUDA.

**Influence of SFA in Transformers.** Figure 3 compares latency of SFA and dense attention across different modular levels of a Transformer: from the raw dot-product to the full model. As sparsity increases (smaller $k$), latency drops significantly. Importantly, the benefit compounds with complexity: while dot-product alone shows modest gains, the full Transformer achieves over 2× reduction. This demonstrates that sparsity scales well when applied throughout the network stack.

Table 3: Evaluation on general reasoning tasks and synthetic retrieval (NIAH). Accuracy is in %.

| Model | Variant | General Tasks | | | NIAH Acc | | | |
|---|---|---|---|---|---|---|---|---|
| | | GSM-8K | Arxiv | PubMed | 4096 | 8192 | 16384 | 32768 |
| Qwen3-0.6B | Base | 59.59 | 13.65 | 10.48 | 90 | 87 | 77 | 52 |
| | Fine-tune | 63.42 | 41.17 | 40.54 | 94 | 92 | 79 | 55 |
| | SFA ($k = 16$) | 61.46 | 39.14 | 39.03 | 95 | 93 | 77 | 53 |
| Qwen3-4B | Base | 75.44 | 31.52 | 29.19 | 97 | 95 | 90 | 81 |
| | Fine-tune | 76.18 | 49.31 | 49.05 | 99 | 96 | 92 | 84 |
| | SFA ($k = 16$) | 75.56 | 46.28 | 47.91 | 99 | 93 | 91 | 84 |
| Qwen3-8B | Base | 87.62 | 40.13 | 37.22 | 100 | 100 | 97 | 92 |
| | Fine-tune | 89.11 | 54.26 | 55.07 | 100 | 100 | 99 | 95 |
| | SFA ($k = 16$) | 87.99 | 52.74 | 52.61 | 100 | 100 | 100 | 97 |

**Influence of Dimension and Context Length.** Figure 4 examines latency under varying head dimensions (128 vs. 256) and context lengths (4k vs. 65k). At shorter contexts (4k tokens), SFA offers consistent but moderate gains. However, under long contexts (65k tokens) and larger head sizes (256 dim), the improvement is dramatic: SFA reduces latency by more than an order of magnitude. This confirms that sparsity is most effective in the large-scale regime, where dense attention becomes prohibitively expensive.

**Latency and Memory Scaling at Inference.** Figure 5 benchmarks autoregressive inference with KV cache. For short contexts ($\leq$4k), dense attention remains competitive because sparse kernels incur lookup overhead. Beyond 8k–16k tokens, however, SFA consistently outperforms dense attention. Moreover, SFA reduces KV-cache size proportionally to sparsity, saving up to $\sim$40% memory at $k = 4$. This makes sparse features especially valuable for long-context inference, where memory footprint is often the limiting factor.

Together, these results show that SFA addresses both compute and memory bottlenecks. During training, it accelerates high-dimension, long-context workloads by cutting FLOPs; during inference, it reduces both latency and KV-cache usage for long sequences. These complementary benefits make SFA well-suited for scaling LLMs to ultra-long contexts. More results are shown in Appendix E.

## 5 EXPLORING SFA ADAPTATION WITH PRETRAINED LLMS

In addition to incorporating SFA during the pretraining stage, we also attempted to adjust models with dense pretraining to a sparse feature attention pattern through fine-tuning. In this section, we explore the use of SFA in fine-tuning.

**Regularized Sparse Finetuning.** During finetuning, we keep SFA consistent with our strategy in the pre-training phase (Eqs. 3 & 6). Nevertheless, the sparsification of pretrained dense features introduces a severe distribution shift for the pretrained model. Therefore, we regularize the finetuning with an additional MSE loss such that SFA's attention scores approximate that of dense features. Since FlashAttention and FlashSPA do not materialize the full attention matrix, in practice we approximate the dense attention output $O_h$ (with stop gradient) with SFA's attention output $\tilde{O}_h$ at each head $h$, leading to the final finetuning objective:

$$\mathcal{L} = \mathcal{L}_{\text{LM}} + \lambda \mathcal{L}_{\text{reg}} = -\mathbb{E}_{(x,y)} \log p_\theta\big(y \mid x; \tilde{S}, V\big) + \lambda \frac{1}{H} \sum_{h=1}^{H} \big\| \tilde{O}_h - \text{stopgrad}(O_h) \big\|_F^2. \quad (8)$$

**Datasets.** To comprehensively evaluate the performance of SFA during fine-tuning, we conduct experiments using mathematical tasks, document question answering, and long-context retrieval tasks. We use GSM-8K (Cobbe et al., 2021), Sci-papers (Arxiv and PubMed (Cohan et al., 2018)), and NIAH data constructed from real texts, respectively. Because applying TopK to the features almost resets the pattern of the previous dense features, we first restore the model's language ability by training on a similar reasoning dataset, MWP-200k (Mitra et al., 2024), before GSM-8K. For the

NIAH data, we use the Pile dataset as haystack for random filling. The size of the training set is set to 100k, and 100 test data entries for each length in the test set.

**Training Settings.** We fine-tune Qwen3-0.6B and Qwen3-4B using Llama-Factory (Zheng et al., 2024) with $k = 16$ for SFA. For mathematical reasoning and science QA tasks, the training context length is set to 16,384 tokens, while for long-context retrieval tasks it is set to 32,768 tokens, with evaluation spanning 4k–32k contexts. All models are trained for three epochs with identical hyperparameters. Detailed experiment setting can be found in Section A.2.

**Result Analysis.** Table 3 compares the base model, dense fine-tuning, and our Top-16 variant. On general tasks, dense fine-tuning yields large gains on Arxiv and PubMed by adapting to the evaluation format, and SFA closely tracks these improvements, showing that sparsified features preserve document-comprehension signals even under hard $k$. On GSM-8K, Top-16 lags slightly behind dense fine-tuning, indicating that arithmetic reasoning is more sensitive to pruning. For long-context retrieval (NIAH), Top-16 performs nearly identically to dense fine-tuning, consistent with Section 4.2, suggesting that sparse supports provide an effective inductive bias for locality. At the 4B scale, Top-16 remains within 1–3 points of dense on general tasks and holds parity on NIAH, confirming its robustness and compatibility with larger backbones.

## 6 RELATED WORK

**Token-level sparsity.** Many approaches reduce the quadratic cost by pruning *which tokens* interact. Structured patterns (local/strided/global) and learned routing yield strong long-context performance: Sparse Transformers (Child et al., 2019), Longformer and BigBird (Beltagy et al., 2020; Zaheer et al., 2020), Routing Transformers (Roy et al., 2021), and Reformer (Kitaev et al., 2020). Recent inference systems dynamically select salient tokens or pages ($H_2O$, SnapKV, Quest) (Zhang et al., 2023; Li et al., 2024; Tang et al., 2024). These methods are orthogonal to ours: they sparsify the *set of tokens*, while we sparsify the *feature coordinates* used to score any retained token pair. In practice, SFA composes with token sparsity and paging by shrinking the per-interaction cost.

**Low-rank/kernel approximations vs. feature sparsity.** A parallel line alters the operator to achieve linear or near-linear time via low-rank or kernel approximations: Linformer projects $K, V$ (Wang et al., 2020); Performer approximates softmax with random features (Choromanski et al., 2021); Nyströmformer uses landmark decompositions (Xiong et al., 2021). These compress information into a dense $r \ll d$ space, often trading expressivity for speed. By contrast, SFA keeps the *high-dimensional* feature space but activates only $k \ll d$ learned coordinates per token; attention scores are computed exactly over the overlap of active supports (no kernel surrogates). This is closer in spirit to sparse coding and sparse embeddings (e.g., SPLADE; CSR) that preserve semantic detail while enabling inverted-index efficiency (Formal et al., 2021; Wen et al., 2025).

**Efficient attention kernels and sparse representations.** FlashAttention reorders computation and IO to keep attention exact while minimizing off-chip traffic (Dao et al., 2022; Dao, 2024; Shah et al., 2024); systems like xFormers and flashinfer expose page/block sparsity primitives (Lefaudeux et al., 2022; Ye et al., 2025). Some works use feature cues to *drive token selection* atop such kernels (e.g., SPAR-Q; LoKI) (Ribar et al., 2024; Singhania et al., 2024). SFA differs by *learning* sparse $Q/K$ codes as first-class representations and introducing an IO-aware kernel (*FlashSFA*) that iterates intersections of active coordinates rather than dense $d$-dimensional products, yielding arithmetic and bandwidth savings proportional to $k$ and composing naturally with token-sparse routing. Our focus is thus complementary: we open the underexplored axis of *feature-level* sparsity inside attention while remaining compatible with token-level sparsity and paging.

## 7 CONCLUSION AND LIMITATIONS

We presented **Sparse Feature Attention (SFA)**, a new approach to scaling long-context Transformers through *dimension-level sparsity*. By learning sparse query/key codes and computing attention via feature overlaps, SFA preserves high-dimensional expressivity while reducing both memory and compute. We introduced two adaptation strategies (end-to-end Top-$k$ finetuning and adapter-based training) and an IO-aware *FlashSFA* kernel that integrates sparsity directly into the online-softmax pipeline. Experiments across synthetic and real tasks show that SFA achieves comparable qual-

ity to dense attention with growing efficiency gains at longer contexts, and complements existing token-level sparsity methods.

While promising, several aspects remain open. Sparse tensor products require stronger support from GPU hardware and CUDA libraries to fully unlock their efficiency, though these system-level challenges are likely to be resolved over time. Very sparse query/key codes can lead to occasional quality degradation, suggesting the need for adaptive sparsity budgets. Finally, how to best combine *token-level* and *dimension-level* sparsity remains an exciting direction, offering the possibility of compounding gains in both compute and memory. We view SFA as a first step toward exploring this new axis of sparsity in attention, and hope it motivates further work at the intersection of representation learning, attention design, and efficient systems.

**Ethics Statement.** This work complies with the ICLR Code of Ethics. Our research primarily utilizes publicly available datasets and pretrained models, and we do not foresee any direct negative societal impacts or ethical concerns arising from our methodology.

**Reproducibility.** We provide detailed descriptions of our methodology, datasets, model configurations, and evaluation metrics in the main text and Appendix. Upon acceptance, we will release source code and scripts to enable full replication of our experiments.

## ACKNOWLEDGEMENTS

Yan Xie, Tiansheng Wen, Tangda Huang, and Bo Chen were supported in part by the National Natural Science Foundation of China under Grant 62576266; in part by the Fundamental Research Funds for the Central Universities QTZX24003 and QTZX23018; in part by the 111 Project under Grant B18039. Yifei Wang and Stefanie Jegelka were supported in part by the NSF AI Institute TILOS (NSF CCF-2112665), and an Alexander von Humboldt Professorship.

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

# A  ADDITIONAL EXPERIMENTAL DETAILS

## A.1  PRETRAINING SETUP

**Model configurations.** Table 4 lists detailed configurations of GPT-2 and Qwen3 models, including parameter counts, hidden dimensions, number of layers/heads, and short-embedding baselines.

| Size | #Parameters | hidden_size | num_layers | num_heads | short_hidden | position_embedding |
|------|-------------|-------------|------------|-----------|--------------|--------------------|
| Small | 124M | 768 | 12 | 12 | 384 | APE |
| Medium | 350M | 1024 | 24 | 16 | 512 | APE |
| Large | 596M | 1024 | 28 | 16/8 | 512 | RoPE |

Table 4: Base model configurations. "Short" refers to halving the hidden size for $Q/K$.

**Implementation notes.** For fairness, short-embedding baselines insert only linear projections before and after attention. For Qwen3, we add an extra linear transformation after RoPE to isolate positional dimensions from sparsification. FlashSFA kernels are used for tiled execution.

**Training.** GPT-2 models are trained on OpenWebText and Qwen3 on The Pile with standard LM objectives. Validation PPL is reported on held-out splits. Zero-shot evaluations follow PiQA, LAMBADA, ARC-e/ARC-c, and HellaSwag. Long-context efficiency is measured as decoding throughput at 128k tokens.

## A.2  FINE-TUNING SETUP

Table 5: Configurations for fine-tuning MoE models

| Model | Dataset | Epoch | Batch_Size | Lr | Warmup_Ratio | Gradient_Ckecpointing |
|-------|---------|-------|------------|------|--------------|----------------------|
| Qwen3-0.6B | GSM8K | 3 | 256 | 6e-4 | 0.1 | False |
| | Arxiv | 2 | 256 | 1e-5 | 0.05 | False |
| | PubMed | 2 | 256 | 2e-5 | 0.05 | False |
| | NIAH | 3 | 256 | 2e-5 | 0.05 | False |
| Qwen3-4B | GSM8K | 3 | 256 | 6e-6 | 0.1 | True |
| | Arxiv | 2 | 256 | 2e-6 | 0.1 | True |
| | PubMed | 2 | 256 | 2e-6 | 0.1 | True |
| | NIAH | 3 | 256 | 2e-6 | 0.1 | True |

# B  ADDITIONAL EXPERIMENTS

## B.1  LATENCY

We benchmarked the latency of the attention module on three feature dimensions: 256, 128, and 64, respectively.

**Prefilling Latency**  The computational complexity of the full attention module can be expressed as $O(n^2 d)$. So we can express Latency as $Latency_{attn} \propto N^2 d$. To better analyze the impact of the feature dimension $d$ on computational complexity, we conduct the analysis in the logarithmic space while fix $Batch = 8$ and $Heads = 8$ :

$$log(Latency_{attn}) \propto 2logN + logd \tag{9}$$

The results in the logarithmic coordinate system are shown in Figure 6. As shown in the figure, we can observe that the latency generally exhibits a linear relationship with the sequence length. Furthermore, the latency gap between different compression ratios are close to a constant value in the logarithmic space, which also indicates that the absolute efficiency improvement achieved by compressing the feature dimension increases exponentially with the sequence length.

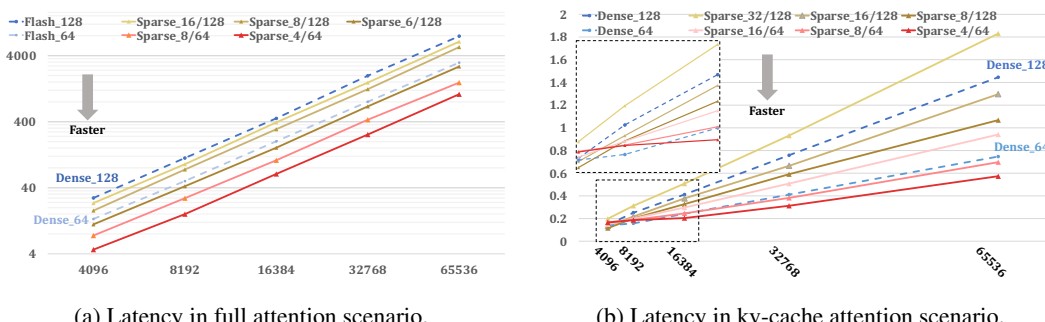

(a) Latency in full attention scenario.    (b) Latency in kv-cache attention scenario.

Figure 6: Comparison of latencies in different attention scenarios: (a) full attention(Time-to-first-token) and (b) kv-cache attention(Time-to-next-token).

**KV-cache Latency** KV cache, which has been widely used in LLM decoding, is known as a memory bound task. Therefore, we benchmarked the inference latency and KV cache memory usage of sparse features and dense features in the KV cache decoding scenario, while keeping $Batch = 8$ and $Heads = 8$ unchanged. Since setting the length of the Query $N_q = 1$, the computational complexity of the decoding attention can be expressed as $O(Nd)$, which means that as the sequence length increases, the computational complexity grows linearly. Our experimental results confirm this. As shown in Table. 2, sparse attention becomes increasingly advantageous as the context length grows. At short sequences (e.g., 4k tokens), dense attention is still competitive or even faster because sparse kernels pay overhead for index lookups and binary searches. However, once the context exceeds about 8k–16k tokens, the sparse variants consistently overtake the dense baselines.

## B.2 FLOPs

To further analyze the operation of full-attention, we separately counted the number of floating-point operations (FLOPs) and integer operations (INOPs) under different settings.

Table 6: **Operation counts for standard flash attention and flash attention sparse.** The number of floating-point operations (FLOPs) and integer operations (INOPs) were counted separately for feature dimensions of 64 and 128 under different context lengths.

| Config | 8192 | | 16384 | | 32768 | | 65536 | |
| --- | --- | --- | --- | --- | --- | --- | --- | --- |
| | **TFLOPs** | **INOPs** | **TFLOPs** | **INOPs** | **TFLOPs** | **INOPs** | **TFLOPs** | **INOPs** |
| Dense_128 | 2.23 | / | 8.92 | / | 35.67 | / | 142.67 | / |
| Sparse_32/128 | 1.20 | 28.31 | 4.79 | 58.72 | 19.17 | 121.63 | 76.70 | 251.67 |
| Sparse_16/128 | 1.15 | 18.87 | 4.59 | 39.85 | 18.35 | 83.89 | 73.40 | 176.17 |
| Sparse_8/128 | 1.13 | 13.63 | 4.54 | 29.36 | 18.14 | 62.91 | 72.57 | 134.22 |
| Dense_64 | 1.12 | / | 4.48 | / | 17.94 | / | 71.75 | / |
| Sparse_16/64 | 0.61 | 15.16 | 2.42 | 29.36 | 9.69 | 60.82 | 38.76 | 125.83 |
| Sparse_8/64 | 0.58 | 9.44 | 2.32 | 19.92 | 9.27 | 41.94 | 37.11 | 88.08 |
| Sparse_4/64 | 0.57 | 6.82 | 2.30 | 14.68 | 9.17 | 31.46 | 36.70 | 67.11 |

As shown in Tab.6, because we directly reduced the number of non-zero elements in the feature vectors, the number of floating-point operations has significantly decreased, and a large proportion of the floating-point operations in the sparse version come from matrix multiplication in the P@V stage. The reason is that sparse feature attention converts a large number of FLOPs into the process of finding overlapping non-zero elements in sparse matrix multiplication, which corresponds to the INOPs in the table.

# C    IMPLEMENTATION DETAILS OF FLASHSFA KERNEL

In this section, we provide a comprehensive breakdown of the FlashSFA CUDA kernel implementation, focusing on the parallelism hierarchy, core data structures, memory access patterns, and a detailed complexity analysis of the sparsification overhead. Since current GPUs do not support general sparse matrix multiplication well, for a fair comparison, we compared Flash Attention Sparse with FMA-based Dense Flash Attention on the code base of Flash Attention 2 in the LeetCUDA open-source library (DefTruth & Others, 2025).

## C.1    ALGORITHM PROCEDURE

---

**Algorithm 1** FlashSFA (forward with tile $(B_r \times B_c)$)

---

**Require:** CSR($\tilde{Q}$): Q_indptr, Q_indices, Q_values; CSC$_{\text{feat}}$($\tilde{K}$): Kf_indptr, Kf_indices, Kf_values; $V$ (dense, row-major in HBM); tile offsets $(i_0, j_0)$; tile sizes $(B_r, B_c)$.

1:  **Init score tile storage:** scores $\leftarrow$ zeros$(B_r, B_c)$ in SRAM.
2:  **Init CSR($P$) row pointers:** P_indptr$[i_0] \leftarrow$ current nnz counter $t_P$.
3:  **for** $r = 0$ **to** $B_r - 1$ **do**                    $\triangleright$ $i = i_0 + r$ is the global query index
4:      $i \leftarrow i_0 + r$
5:      **Register accumulator:** $row\_scores[0{:}B_c] \leftarrow 0$        $\triangleright$ kept in registers per thread/warp
6:      $t_L \leftarrow Q\_indptr[i];\ \ t_R \leftarrow Q\_indptr[i+1]$
7:      **for** $t = t_L$ **to** $t_R - 1$ **do**                    $\triangleright$ iterate nonzeros of query row $i$
8:          $f \leftarrow Q\_indices[t];\ \ qv \leftarrow Q\_values[t]$
9:          $p_0 \leftarrow Kf\_indptr[f];\ \ p_1 \leftarrow Kf\_indptr[f+1]$            $\triangleright$ posting list for feature $f$
10:         $(p_L, p_R) \leftarrow \text{BINARYSEARCHRANGE}\big(Kf\_indices[p_0{:}p_1], [j_0, j_0{+}B_c)\big)$
11:         **for** $p = p_L$ **to** $p_R - 1$ **do**                    $\triangleright$ only keys $j$ that fall inside the key tile
12:             $j \leftarrow Kf\_indices[p];\ \ c \leftarrow j - j_0$
13:             $kv \leftarrow Kf\_values[p]$
14:             $row\_scores[c] \mathrel{+}= (qv \cdot kv)/\sqrt{d}$        $\triangleright$ feature-overlap accumulation in registers
15:         **end for**
16:     **end for**
17:     **for** $c = 0$ **to** $B_c - 1$ **do**
18:         $scores[r, c] \leftarrow row\_scores[c]$        $\triangleright$ store to SRAM after register accumulation
19:     **end for**
20: **end for**
21: **Mask (optional):** apply causal mask in-place to scores.
22: **Online softmax per row (as in FA)**
23: **for** $r = 0$ **to** $B_r - 1$ **do**
24:     $i \leftarrow i_0 + r$
25:     $\mathbf{o}_i \leftarrow$ zeros$(d_v)$ in registers                    $\triangleright$ accumulator for output row $i$
26:     $t_L \leftarrow P\_indptr[i];\ \ t_R \leftarrow P\_indptr[i+1]$
27:     **for** $t = t_L$ **to** $t_R - 1$ **do**                    $\triangleright$ iterate nonzeros $P_{ij}$ in row $i$
28:         $j \leftarrow P\_indices[t];\ \ p \leftarrow P\_values[t]$
29:         $\mathbf{v}_j \leftarrow V[j, 0{:}d_v]$        $\triangleright$ row-vector, contiguous load from HBM
30:         $\mathbf{o}_i \mathrel{+}= p \cdot \mathbf{v}_j$
31:     **end for**
32:     **Write back:** add $\mathbf{o}_i$ to the corresponding row of $O$.
33: **end for**

---

**Binary Search for Tiling.**    To restrict computations to the current processing tile $[j_0, j_0 + B_c)$, we employ a `BINARY_SEARCH_RANGE` routine.  for a feature $f$, we search the sorted array `Kf_indices` to find the sub-range $[p_{lo}, p_{hi})$ such that all indices fall within the current key tile. Since this operation runs in registers with a fixed number of iterations, it is highly efficient and branch-regular.

## C.2 PARALLELISM MODEL

The FlashSFA kernel adopts a tiling strategy similar to standard FlashAttention but optimized for sparse operations. The mapping of CUDA threads, warps, and blocks to the computation is designed to maximize occupancy and eliminate the need for atomic operations during score accumulation.

**Grid and Block Mapping.** The computation grid is defined as $grid = (\lceil N/B_r \rceil, B \times H)$, where $N$ is the sequence length, $B_r$ is the row tile height, $B$ is the batch size, and $H$ is the number of heads.

- **Grid:** `blockIdx.x` selects a row tile of height $B_r$ (typically 128), and `blockIdx.y` selects the specific batch-head pair.
- **Block:** Each thread block consists of 256 threads (8 warps). A single block is responsible for computing the attention output for a specific row tile $[r_0, r_0 + B_r)$.

**Warp and Thread Hierarchy.** Within a block, the workload is distributed as follows:

- **Warps:** The 8 warps in a block process disjoint stripes of rows. Each warp is assigned a 16-row stripe within the $B_r$ rows handled by the block.
- **Threads:** Within a warp, threads are mapped to a 2D grid to process the score matrix. Each thread is responsible for a $2 \times 2$ patch of the score tile (two rows × two columns). Across all 32 threads in a warp, these patches perfectly tile the warp's assigned stripe.

**Loop Parallelization and Atomicity.** The kernel iterates through column tiles (Key/Value blocks) in the outer loop. Inside the loop:

- For the Query ($Q$), lines are distributed across warps.
- For the non-zero elements (nnz) within a $Q$ row, threads iterate sequentially over the CSR segment.

**No Atomic Operations.** A critical design choice is the absence of atomic operations (e.g., `atomicAdd`) for score accumulation. Since each output score position $S[r, c]$ in the tile is owned by exactly one thread (via the fixed $2 \times 2$ patch mapping), each thread accumulates all partial contributions for its assigned scores in registers. This ensures thread safety and maximizes throughput.

## C.3 CORE DATA STRUCTURES

FlashSFA relies on a specialized sparse storage format to efficiently handle the intersection of active queries and keys.

**Feature-wise CSC Format** ($CSC_{feat}$). Standard sparse matrices typically store data in Compressed Sparse Row (CSR) or Column (CSC) format where dimensions correspond to token indices. However, to efficiently retrieve relevant keys for a given active feature in the query, we utilize a transposed view for the Key matrix $K$, denoted as $CSC_{feat}(\bar{K})$.

- **Columns:** Correspond to *Feature IDs* ($f \in [0, d)$).
- **Rows:** Correspond to *Key IDs* ($j \in [0, N)$).

This layout allows the kernel to quickly access the posting list (list of token indices) for any specific feature $f$ activated by the query.

## C.4 MEMORY ACCESS STRATEGY

Efficient memory access is paramount for GPU performance. FlashSFA employs specific strategies to ensure coalesced access despite the irregularity of sparse data.

**Accessing Q (CSR).** For the Query matrix stored in CSR format, each block processes a contiguous row range $[r_0, r_0 + B_r)$. We calculate the span of non-zero elements $[q_{lo}, q_{hi}) = [Q_{indptr}[r_0], Q_{indptr}[r_0 + B_r])$, which corresponds to a single contiguous segment in HBM. This allows for streamlined streaming of $Q$ indices and values.

**Accessing K (Coalesced Sparse Reads).** For a given key tile and active feature $f$, once the sub-range $[p_{lo}, p_{hi})$ is identified via binary search:

- Warps cooperatively load the indices and values from $CSC_{feat}$.
- We utilize lane-strided access patterns, e.g., loading index $p = p_{lo} + \texttt{lane\_id} + 32 \times k$.

This ensures that even though the semantic access is sparse (random features), the physical memory transactions are coalesced into contiguous slices of HBM, which are then staged into shared memory.

**Accessing V (Sparse-Dense Multiplication).** The computation of $O = P@V$ involves a sparse attention matrix $P$ and a dense value matrix $V$. Since $V$ is stored in a row-major dense layout, the access to any specific row $V_j$ is contiguous:

- The non-zero structure of $P$ determines which rows of $V$ to access.
- Threads in a warp cooperatively load row $V_j$ using vectorized instructions (e.g., `float4` or `half2`), ensuring high bandwidth utilization.

**Memory Bandwidth Analysis** Empirical profiling results in Table 7 confirms that the memory system delivers close to peak bandwidth (approx. 919 GB/s in "memory-only" benchmarks), indicating that memory access to $V$ is not a bottleneck.

Table 7: HBM Bandwidth Comparison. "w/o compute" denotes measuring memory throughput with computation logic disabled.

| Kernel | Dense | Dense w/o compute | FlashSFA | FlashSFA w/o compute |
|---|---|---|---|---|
| **HBM Bandwidth (GB/s)** | 14.22 | 1194.34 | 17.14 | 919.38 |

## C.5 REVISED COMPLEXITY ANALYSIS

A potential concern with sparse attention is the overhead introduced by the sparsification process (Top-$k$ selection). SFA utilizes RTop-$k$ kernel(Xie et al., 2024) to sparsify $Q$ and $K$ with a computational complexity of $O(Nd)$. This kernel employs GPU-parallelized binary search where each warp processes a feature row. As shown in Table 8, the latency of RTop-$k$ is negligible compared to the full attention computation.

Table 8: **Latency comparison (ms) between standard `torch.topk` and RTop-$k$ kernel across different context lengths.** The Ratio indicates the percentage of time RTop-$k$ consumes relative to the total attention forward pass.

| Context Length ($N$) | 1024 | 4096 | 8192 | 16384 | 32768 | 65536 |
|---|---|---|---|---|---|---|
| `torch.topk` | 0.730 | 2.701 | 5.336 | 10.684 | 21.205 | 42.374 |
| RTop-$k$ (Ours) | 0.221 | 0.589 | 1.089 | 2.080 | 4.057 | 8.080 |
| *Ratio of RTop-k (%)* | *10.51* | *2.10* | *1.96* | *1.90* | *1.03* | *0.51* |

**Complexity of $P@V$.** The multiplication $P@V$ effectively becomes a Sparse Matrix-Matrix Multiplication (SpMM). Since the sparsity of $P$ is induced by the sparsity of $Q$ and $K$, the number of non-zero elements in $P$ is significantly reduced. While a standard dense attention requires $O(N^2d)$ operations, SFA reduces the effective FLOPs count significantly. The memory access pattern described in Section C.3 ensures that the theoretical FLOPs reduction translates into actual wall-clock speedup by maintaining high memory bandwidth efficiency.

# D   LATENCY TIMING RESULTS

Table 9: Latency (ms) versus context length.

| | Context Length | | | | | |
|---|---|---|---|---|---|---|
| **Variant** | **1024** | **4096** | **8192** | **16384** | **32768** | **65536** |
| Dense_256 | 10.98 | 176.20 | 712.98 | 2894.46 | 11 772.47 | 49 197.70 |
| Sparse_32/256 | 3.21 | 45.89 | 180.38 | 715.02 | 2886.21 | 11 529.74 |
| Sparse_24/256 | 2.77 | 33.96 | 154.79 | 612.81 | 2488.04 | 8309.51 |
| Sparse_16/256 | 2.31 | 29.15 | 128.52 | 510.18 | 2079.82 | 7388.78 |
| Sparse_12/256 | 1.97 | 21.28 | 109.04 | 431.50 | 1769.73 | 5063.70 |
| Sparse_10/256 | 1.93 | 20.06 | 106.93 | 422.95 | 1734.99 | 4877.05 |
| Sparse_8/256 | 1.86 | 17.41 | 102.63 | 405.52 | 1665.69 | 4235.00 |
| Sparse_6/256 | 1.68 | 15.62 | 96.25 | 365.26 | 1505.59 | 3841.40 |
| Sparse_4/256 | 1.51 | 13.94 | 82.48 | 324.97 | 1345.26 | 3412.10 |
| Sparse_2/256 | 1.43 | 13.32 | 77.66 | 305.81 | 1273.37 | 2999.38 |
| Dense_128 | 2.10 | 28.05 | 112.88 | 449.61 | 1981.92 | 7879.33 |
| Sparse_32/128 | 2.17 | 31.01 | 120.58 | 465.68 | 1802.03 | 7101.95 |
| Sparse_28/128 | 1.80 | 25.06 | 98.13 | 387.02 | 1535.46 | 6103.98 |
| Sparse_24/128 | 1.70 | 23.84 | 94.10 | 373.42 | 1486.14 | 5909.66 |
| Sparse_16/128 | 1.56 | 17.94 | 70.62 | 279.72 | 1108.96 | 4412.02 |
| Sparse_12/128 | 1.11 | 16.25 | 64.39 | 255.24 | 1017.04 | 4047.53 |
| Sparse_10/128 | 1.00 | 15.03 | 60.41 | 239.77 | 954.21 | 3814.06 |
| Sparse_8/128 | 0.92 | 13.17 | 54.25 | 215.37 | 777.15 | 3323.53 |
| Sparse_6/128 | 0.79 | 11.17 | 42.24 | 161.43 | 681.73 | 2738.84 |
| Sparse_4/128 | 0.67 | 10.08 | 40.09 | 157.88 | 579.75 | 2576.93 |
| Sparse_2/128 | 0.58 | 9.90 | 38.92 | 154.71 | 539.49 | 2423.82 |
| Dense_64 | 0.77 | 13.51 | 50.62 | 202.56 | 801.50 | 3137.78 |
| Sparse_16/64 | 0.90 | 12.51 | 39.41 | 195.53 | 779.19 | 2963.94 |
| Sparse_12/64 | 0.70 | 9.71 | 38.23 | 151.60 | 603.18 | 2400.37 |
| Sparse_10/64 | 0.67 | 9.23 | 36.36 | 144.17 | 573.31 | 2282.26 |
| Sparse_8/64 | 0.59 | 8.14 | 32.00 | 126.99 | 504.23 | 2014.14 |
| Sparse_6/64 | 0.51 | 7.05 | 27.64 | 109.36 | 434.58 | 1727.43 |
| Sparse_4/64 | 0.41 | 5.41 | 21.07 | 83.12 | 328.83 | 1311.59 |
| Sparse_2/64 | 0.39 | 5.15 | 19.75 | 77.96 | 309.13 | 1233.64 |

# E    COMPARISON OF EFFICIENT ATTENTION BY TRAINING

Table 10: **Latency, Perplexity and Accuracy results** comparison with various compression and acceleration techniques, categorized into *Token-Level* and *Feature-Level* Operations. For token-level operations, "Longforemer" (Beltagy et al., 2020) denotes fixed token sparsity pattern, "NSA" (Yuan et al., 2025) denotes dynamic token sparsity pattern. "Dense (full)" baselines use full hidden size and uncompressed KV cache; "Short ($d = X$)" denotes baselines with half feature dimensions; "Quant" denotes 8-bit quantization aware training (QAT (Liu et al., 2024b)) on weights and activations; "Low-Rank" denotes PCA-based projection matrix fine-tuning; "MLA" denotes multi head latent attention (Liu et al., 2024a), and "MLA + SFA" combines SFA with latent key/value. "Latency@128k" is measured by "Decoding with KV cache (TTNT) (ms)" and "Prefilling with full attention (TTFT) (s)". PPL is evaluated on OpenWebText for GPT-2 and Pile for Qwen3.

| Model | Variant | Latency@128k ↓ | | PPL ↓ | Acc ↑ | | | | | |
| | | Decode | Forward | OWT/Pile | PiQA | LAMBDA | ARC-e | ARC-c | HellaS | Avg |
|---|---|---|---|---|---|---|---|---|---|---|
| GPT2 124M | Dense (full) | 17.08 | 16.86 | 17.29 | 56.34 | 22.78 | 28.35 | 14.32 | 19.61 | 28.28 |
| | | | | Token-Level Operation | | | | | | |
| | Longformer | 6.75 | 7.93 | 18.73 | 54.25 | 21.27 | 28.02 | 13.01 | 18.92 | 28.10 |
| | +SFA ($k = 8$) | 5.23 | 6.18 | 19.30 | 52.81 | 20.54 | 26.39 | 12.59 | 17.24 | 25.91 |
| | | | | Feature-Level Operation | | | | | | |
| | Short ($d = 32$) | 8.37 | 7.86 | 20.70 | 51.30 | 19.39 | 25.72 | 12.47 | 14.26 | 24.63 |
| | Low-Rank | 8.93 | 7.99 | 19.89 | 51.79 | 20.04 | 26.47 | 12.92 | 14.99 | 25.24 |
| | MLA | 5.04 | 15.39 | 17.38 | 57.83 | 22.29 | 28.37 | 13.92 | 19.66 | 28.41 |
| | MLA + SFA | 3.98 | 15.05 | 19.07 | 54.33 | 21.92 | 27.88 | 13.10 | 19.01 | 27.25 |
| | Quant | 14.26 | 12.97 | 17.64 | 56.18 | 21.03 | 28.09 | 13.58 | 19.05 | 27.59 |
| | SFA ($k = 8$) | 14.12 | 9.41 | 18.17 | 54.92 | 21.03 | 28.41 | 13.41 | 19.26 | 27.40 |
| | SFA (quant) | 12.28 | 8.72 | 18.54 | 54.53 | 20.81 | 28.39 | 13.27 | 18.97 | 27.12 |
| Qwen3 0.6B | Dense(full) | 80.84 | 77.65 | 4.66 | 62.47 | 34.82 | 45.41 | 20.35 | 33.95 | 39.40 |
| | | | | Token-Level Operation | | | | | | |
| | NSA | 9.73 | 20.32 | 4.57 | 62.69 | 35.01 | 45.10 | 20.47 | 34.42 | 39.54 |
| | +SFA ($k = 16$) | 8.85 | 17.17 | 4.95 | 60.02 | 33.58 | 42.74 | 18.31 | 32.48 | 37.43 |
| | | | | Feature-Level Operation | | | | | | |
| | Short ($d = 64$) | 38.68 | 30.84 | 6.03 | 58.43 | 31.27 | 41.58 | 15.83 | 28.29 | 35.08 |
| | Low-Rank | 40.58 | 32.46 | 5.50 | 59.19 | 31.49 | 41.77 | 15.80 | 30.65 | 35.78 |
| | MLA | 8.74 | 68.92 | 4.69 | 62.39 | 34.71 | 45.41 | 20.17 | 34.21 | 39.38 |
| | MLA + SFA | 6.72 | 65.29 | 4.9 | 61.22 | 33.94 | 43.36 | 19.25 | 33.94 | 38.34 |
| | Quant | 72.23 | 59.73 | 4.71 | 62.29 | 34.33 | 45.39 | 20.02 | 33.91 | 39.19 |
| | SFA ($k = 16$) | 66.29 | 34.20 | 4.81 | 61.73 | 34.05 | 45.62 | 19.27 | 34.03 | 38.94 |
| | SFA (quant) | 57.47 | 30.74 | 5.16 | 59.63 | 33.10 | 44.93 | 15.98 | 33.64 | 37.46 |

Table 10 compares SFA with a variety of token-level and feature-level compression / acceleration techniques on GPT-2 124M and Qwen3-0.6B. We report both prefill ("Forward") and decoding ("Decode") latency at 128K context, together with perplexity and downstream accuracy.

**Orthogonality to token-level methods.** For token-level operations, SFA is applied on top of Longformer and NSA as a drop-in replacement for their dense attention blocks. In both models, adding SFA consistently reduces both Decode and Forward latency while achieving comparable performance. This shows that SFA is orthogonal to token-level sparsification: it can be combined with existing token-level sparse attention methods to further accelerate long-context inference.

**Feature-level speed–accuracy trade-off.** Among feature-level methods, SFA can also be combined with MLA(on the compressed latent vector) and quantization. Pure SFA reduces latency compared to the dense baseline while keeping PPL and average accuracy close. Compared with Short and Low-Rank feature compression, which suffer larger accuracy drops, SFA and SFA (quant) maintain much higher accuracy at similar or better speed. Overall, SFA and its combinations deliver the strongest performance among feature-level approaches while still providing significant end-to-end speedups.

# F  LOAD BALANCE

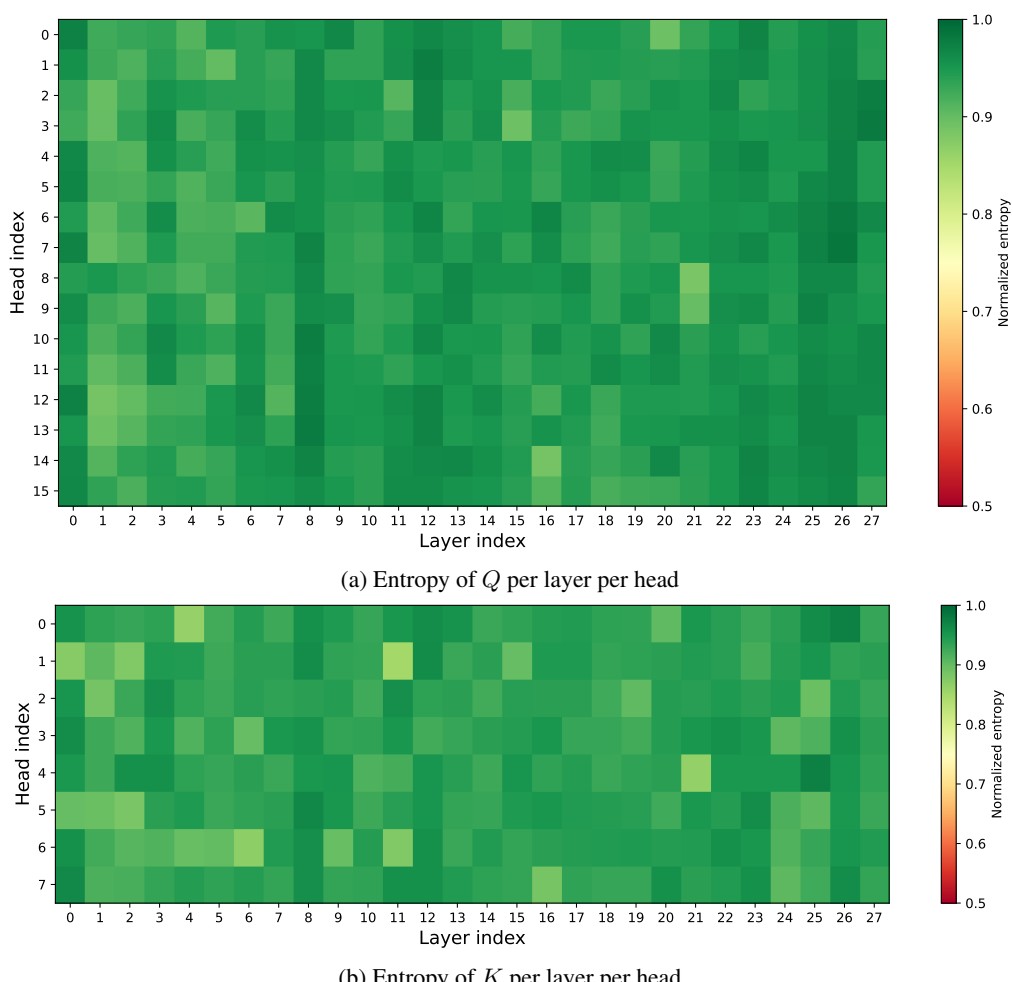

(a) Entropy of $Q$ per layer per head

(b) Entropy of $K$ per layer per head

Figure 7: **Entropy of Top-K feature selection across layers and heads.** We plot the normalized entropy of the TopK index distribution for each attention head and layer of Qwen3-0.6B when applying SFA. (a) Entropy of the TopK positions of query vectors $Q$ (16 heads, due to GQA). (b) Entropy of the TopK positions of key vectors $K$ (8 heads). Each cell corresponds to one (layer, head) pair, and **brighter colors indicate higher entropy (more balanced use of feature dimensions)**.

A natural concern for TopK sparsification on $Q$ and $K$ is that some heads or layers might collapse to using only a few feature dimensions, leading to poor load balance. To study this, we measure the normalized entropy of the TopK index distribution for every head and layer on a small but diverse evaluation set: we sample 50 samples from each of the Arxiv, Github, FreeLaw, and PubMed domains in the Pile validation split (200 samples in total) (Figure 7). **For the 16 $Q$-heads in Qwen3 (due to GQA), the entropy ranges from 0.88 to 0.98 with an average of 0.94. For the 8 $K$-heads, the entropy ranges from 0.85 to 0.97 with an average of 0.93.** These values are close to the maximum possible entropy (1.0) and show only mild variation across layers and heads, indicating that the selected TopK dimensions remain well distributed rather than concentrating on a few indices.

Although SFA does **not** introduce any explicit load-balance loss, the feature activations remain nearly balanced. We hypothesize that, unlike TopK applied to *weights* (as in MoE routing), applying TopK directly on **feature vectors** during end-to-end training encourages the model to exploit its full expressive capacity: different dimensions are naturally used whenever they help reduce the training objective. As a result, the model tends to learn a near-uniform utilization of features.

# G    Orthogonal Baselines

Table 11: **Comparison results with token-sparse, KV-pruning, low-rank, and kernel baselines on GPT-2.** Token-sparse training methods include Routing(Roy et al., 2021) and Longformer(Beltagy et al., 2020); KV-pruning (training-free) methods include H$_2$O(Zhang et al., 2023), Quest(Tang et al., 2024), and SnapKV(Li et al., 2024); Loki(Singhania et al., 2024) is a low-rank key compression method (training-free); Performer(Choromanski et al., 2021) is a kernel-based approximation. Rows marked "+SFA ($k = 8$)" apply our feature-sparse SFA to Longformer and SnapKV, showing that SFA is orthogonal to these approaches and can be combined with them.

| Model | | Latency@128k | | PPL | Acc | | | | | |
|---|---|---|---|---|---|---|---|---|---|---|
| | | Decode | Forward | OWT/Pile | PiQA | LAMBDA | ARC-e | ARC-c | HellaS | Avg |
| GPT2 124M | Dense (full) | 17.08 | 16.86 | 17.29 | 42.74 | 22.78 | 28.35 | 8.12 | 19.61 | 24.32 |
| | SFA | 14.12 | 9.41 | 18.17 | 41.62 | 21.03 | 28.41 | 7.39 | 19.26 | 23.54 |
| | Token Sparse (Training) | | | | | | | | | |
| | Routing | 7.92 | 8.37 | 18.64 | 41.39 | 21.08 | 28.31 | 7.11 | 18.89 | 23.35 |
| | Longformer | 6.75 | 7.93 | 18.73 | 41.28 | 21.27 | 28.02 | 7.01 | 18.92 | 23.30 |
| | +SFA ($k = 8$) | 5.23 | 6.18 | 19.30 | 40.75 | 20.54 | 26.39 | 6.63 | 17.24 | 22.31 |
| | KV-pruning (Training-free) | | | | | | | | | |
| | H$_2$O | 13.32 | 16.86 | 18.02 | 41.81 | 20.55 | 27.04 | 7.38 | 18.75 | 23.11 |
| | Quest | 10.84 | 16.86 | 17.95 | 42.34 | 20.79 | 28.3 | 7.82 | 18.83 | 23.62 |
| | SnapKV | 9.88 | 16.86 | 17.91 | 42.49 | 21.92 | 28.43 | 8.01 | 19.38 | 24.05 |
| | +SFA ($k = 8$) | 6.92 | 9.41 | 19.44 | 39.99 | 20.24 | 27.13 | 6.83 | 17.74 | 22.39 |
| | Low-rank keys (Training-free) | | | | | | | | | |
| | Loki | 11.39 | 16.86 | 17.82 | 42.1 | 21.29 | 28.01 | 7.99 | 19.24 | 23.73 |
| | +SFA ($k = 8$) | 9.09 | 9.41 | 19.29 | 40.83 | 20.04 | 27.85 | 7.13 | 18.03 | 22.78 |
| | Kernel Method | | | | | | | | | |
| | Performer | 9.43 | 7.93 | 19.72 | 39.83 | 19.11 | 26.72 | 6.77 | 15.38 | 21.56 |

**Orthogonality and composability with existing token sparse methods.** Table 11 compares SFA with representative long-context techniques on GPT-2 124M and Table 10 compares SFA with other efficient attention methods. As a standalone replacement of dense attention, SFA already improves efficiency over the dense baseline while perplexity and average accuracy remain close. More importantly, **SFA is orthogonal to existing methods and can be combined with them for additional gains.**

**Token-sparse training methods.** When applied on top of Longformer, we sparsify selected tokens. SFA further reduces latency from 6.75/7.93 to 5.23/6.18 ($\approx 1.3\times$ faster decode and prefill), with only a modest change in quality. This shows that feature-level sparsification in SFA complements token-level sparsity patterns.

**KV-pruning and Low-rank keys methods**. KV-pruning methods such as H$_2$O, Quest, and SnapKV improve speed by compressing the number of tokens in the KV cache, so they only accelerate the **Decode** stage and leave **Forward** latency unchanged. When we combine SFA with SnapKV, we obtain additional acceleration in both stages. Similar behavior holds relative to H$_2$O and Quest. This shows that SFA is complementary to KV-pruning: KV-pruning reduces the number of cached tokens for decoding, while SFA sparsifies feature dimensions and brings **additional** gains.

# H ABLATION

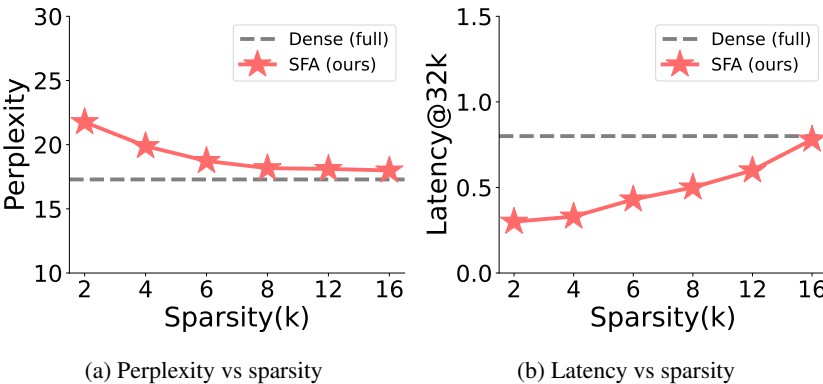

(a) Perplexity vs sparsity        (b) Latency vs sparsity

Figure 8: **Ablation of sparsity $k$ on GPT-2 124M with fixed head dimension $d = 64$.** Perplexity on OpenWebText (left) and latency at 32K context (right) as a function of the Top-$k$ sparsity level used by SFA. The dashed gray line denotes the dense (full) attention baseline; the red curve shows SFA with different $k$.

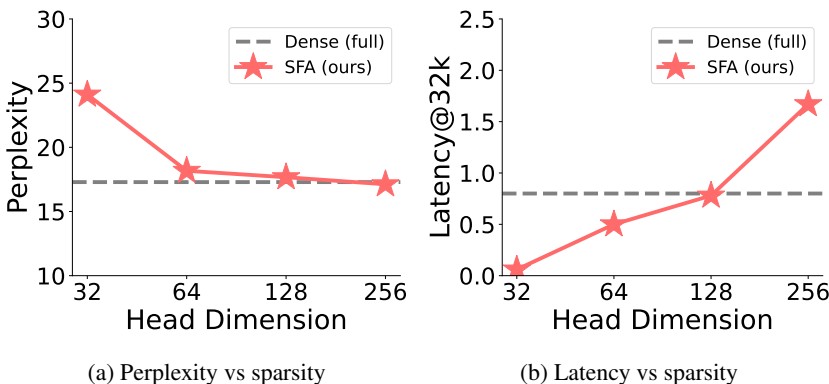

(a) Perplexity vs sparsity        (b) Latency vs sparsity

Figure 9: **Ablation of head dimension $d_{\text{head}}$ on GPT-2 124M with fixed sparsity $k = 8$.** Perplexity on OpenWebText (left) and latency at 32K context (right) as a function of the head dimension $d_{\text{head}}$ used by SFA. The dashed gray line denotes the dense (full) attention baseline; the red curve shows SFA with different $d_{\text{head}}$.

**Sensitivity to sparsity $k$.** Figure 8 studies how the Top-$k$ sparsity level affects performance. As $k$ increases from very sparse settings (e.g., $k = 2$) to denser ones (e.g., $k = 16$), perplexity monotonically decreases and quickly approaches the dense baseline; for $k \geq 8$, the SFA curve is very close to dense attention. In contrast, latency at 32K grows smoothly with $k$: very small $k$ yields the largest speedup, **while moderate $k$ (around $k = 8$) still keeps a substantial latency advantage over the dense model with only a small perplexity gap.** Overall, SFA exhibits a stable speed–accuracy trade-off and is not overly sensitive to the exact choice of $k$, allowing practitioners to pick $k$ to match a desired latency budget.

**Sensitivity to head dimension $d_{\text{head}}$.** Figure 9 varies the head dimension while keeping SFA enabled. When the heads are extremely small (e.g., $d_{\text{head}} = 32$), perplexity degrades noticeably. As we increase the dimension, perplexity quickly improves, and at $d_{\text{head}} = 64$ it is already very close to the dense baseline while latency remains substantially lower. Further increasing $d_{\text{head}}$ beyond 64 brings only marginal perplexity gains but steadily increases latency. Thus $d_{\text{head}} = 64$ emerges as the sweet spot of the speed–accuracy trade-off: it recovers most of the dense-model performance while preserving most of the acceleration provided by SFA.

## I TRAINING STABILITY ANALYSIS

In this section, we investigate the training process of SFA. Figure 10 illustrates the validation loss trajectories across varying sparsity levels ($k \in \{2, 4, 8, 16\}$) of GPT2-124M. We observe that the loss curves exhibit smooth, monotonic convergence devoid of divergent spikes or chaotic oscillations. Notably, even under the most aggressive sparsity constraint ($k = 2$, red line), the model converges steadily. These empirical results suggest that SFA can intrinsically maintain training stability without suffering from excessive variance or optimization instability.

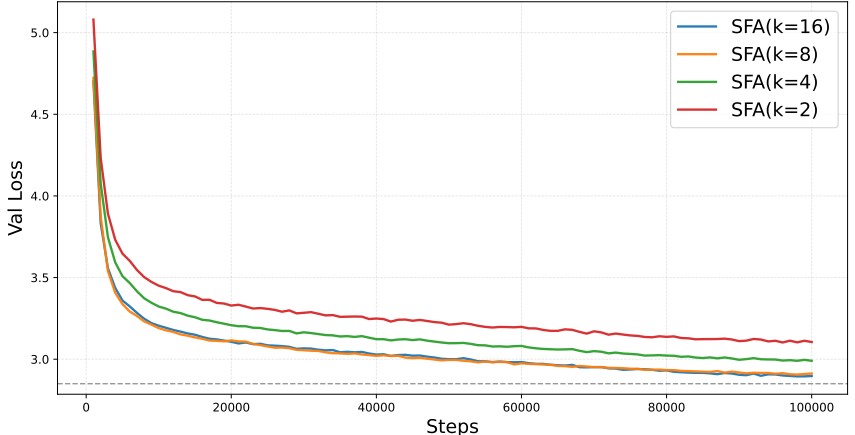

Figure 10: **Validation loss curves of SFA on GPT-2 (124M) pre-training.** We compare varying sparsity levels $k \in \{2, 4, 8, 16\}$. The curves decrease smoothly and monotonically without divergent spikes, demonstrating that SFA maintains training stability even under aggressive sparsity ($k = 2$).

## J MEMORY SAVING

In our implementation, **memory gain can be achieved when** $k < \frac{2}{3}d$. The memory savings of SFA compared to the dense model depend on the data precision used to store the `col_indices` array and `row_pointer` array in the CSR matrix.

For a CSR matrix with shape $(N, d)$ where each row has a fixed number of $k$ non-zero values, the required bytes for each component are calculated as follows:

- **`value` array memory**:
$$\text{Mem}_{\text{value}} = (N \times k) \times S_{\text{val}} \tag{10}$$

- **`indices` array memory**:
$$\text{Mem}_{\text{indices}} = (N \times k) \times S_{\text{idx}} \tag{11}$$

- **`indptr` array memory** (length is $N + 1$):
$$\text{Mem}_{\text{indptr}} = (N + 1) \times S_{\text{ptr}} \tag{12}$$

Where $S$ denotes the number of bytes for the data format.

Therefore, the total memory consumption of the CSR format is the sum of these three parts:

$$\text{Mem}_{\text{csr}} = \text{Mem}_{\text{value}} + \text{Mem}_{\text{indices}} + \text{Mem}_{\text{indptr}} \tag{13}$$

Substituting the above formulas, we obtain the final memory consumption formula:

$$\begin{aligned} \text{Mem}_{\text{csr}} &= (N \times k \times S_{\text{val}}) + (N \times k \times S_{\text{idx}}) + ((N+1) \times S_{\text{ptr}}) \\ &= N \times k \times (S_{\text{val}} + S_{\text{idx}}) + (N+1) \times S_{\text{ptr}} \end{aligned} \tag{14}$$

Consequently, compared to a dense matrix of the same shape, the ratio of memory consumption is:

$$\text{Ratio} = \frac{\text{Mem}_{\text{dense}}}{\text{Mem}_{\text{csr}}} = \frac{N \times d \times S_{\text{val}}}{N \times k \times (S_{\text{val}} + S_{\text{idx}}) + (N+1) \times S_{\text{ptr}}} \approx \frac{d \times S_{\text{val}}}{k \times (S_{\text{val}} + S_{\text{idx}}) + S_{\text{ptr}}} \tag{15}$$

As the $Q/K$ feature dimension in Transformers is generally small, `indices` are typically stored in `int8` format and `indptr` in `int32` format. When we use `fp16`/`bf16` to store the `value` array:

$$\text{Ratio} = \frac{\text{Mem}_{\text{dense}}}{\text{Mem}_{\text{csr}}} \approx \frac{d \times 2}{k \times (2+1) + 4} = \frac{2d}{3k + 4} \approx \frac{2d}{3k} \tag{16}$$

## K  ADDITIONAL NIAH EXPERIMENT

To verify that SFA functions effectively as a general-purpose mechanism without requiring task-specific supervision, we evaluated the retrieval capabilities of SFA in a zero-shot setting. We trained the Qwen3-0.6B model equipped with SFA solely on general language corpora (standard pre-training) and evaluated it on the NIAH task.

As presented in Table 12, SFA consistently outperforms the dense attention baseline across all tested context lengths (1k to 4k), despite lacking specific training for retrieval tasks.

At a context length of 4k, SFA ($k = 16$) achieves an accuracy of **71%**, significantly surpassing the dense baseline (62%). Even with aggressive sparsity ($k = 8$), SFA maintains superior performance (66%).

In addition to improved accuracy, SFA provides substantial speedups. Specifically, SFA ($k = 8$) achieves a **1.5× speedup** at 4k context length compared to the dense baseline.

These findings indicate that feature-level sparsification does not introduce an information bottleneck. On the contrary, the results suggest that SFA preserves essential semantic information while potentially filtering out noise in long-context scenarios, allowing it to function effectively within a general-purpose foundation model paradigm.

Table 12: **NIAH accuracy (%) within 4k Context Length.** Qwen3-0.6B trained on Pile dataset with 4k window, and the accuracy rate on NIAH test lengths from 1k to 4k.

| Context Length | 1k | 2k | 3k | 4k | Speedup@4k |
|---|---|---|---|---|---|
| Dense(full) | 93 | 87 | 79 | 62 | 1.0x |
| SFA($k = 8$) | 95 | 90 | 80 | 66 | **1.5x** |
| SFA($k = 16$) | **96** | **90** | **83** | **71** | 1.2x |

# L  SVD ANALYSIS

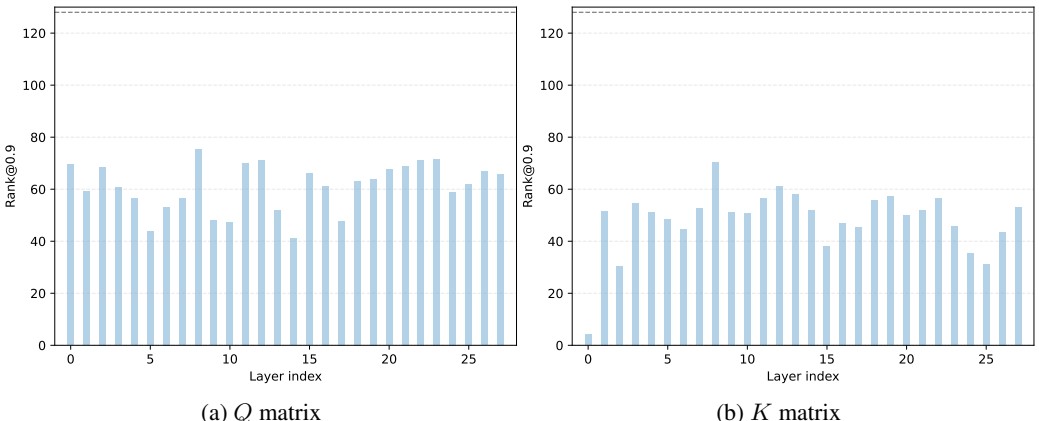

(a) $Q$ matrix                    (b) $K$ matrix

Figure 11: **Eigenvalue spectrum analysis for Qwen3-0.6B model.** Layer-wise effective dimension of (a) query and (b) key activation with normalized cumulative eigenvalue of 0.9, evaluated on the same sampled subset of the Pile validation set in Appendix F.

To better understand why Top-$k$ feature sparsification can preserve semantic information in attention, we analyze the intrinsic dimensionality of the query and key representations in pretrained dense model.

We use the pretrained Qwen3-0.6B model and run it on the same sampled subset of the Pile validation set in Appendix F. For each transformer layer and attention head, we collect the corresponding query and key vectors $Q, K \in \mathbb{R}^d$ (with head dimension $d = 128$). We then perform singular value decomposition (SVD) on the stacked feature matrices and compute the *effective rank* at a given energy threshold $\tau = 0.9$.

As shown in Figure 11, despite the nominal head dimension $d = 128$, both $Q$ and $K$ exhibit consistently low effective rank, typically around 50–60 across layers. This confirms that the attention features lie on a low-dimensional manifold and are therefore *highly compressible*. The key matrices tend to have slightly lower effective rank than queries, but both are far from full rank, indicating substantial redundancy in the dense representations.

# M  LLM USAGE STATEMENT.

In line with the ICLR policy, we disclose the use of Large Language Models during the preparation of this manuscript. Our use of these tools was strictly limited to assistance with language and formatting. Specifically, we employed an LLM to correct grammatical errors and improve the clarity and readability of sentences. The LLM had no role in the core scientific aspects of this work, including research ideation, methodological design, experimental analysis, or the generation of any results or conclusions. All intellectual contributions and the core content of this paper are solely the work of the authors.

