# OpenReview forum: "Scaling Attention via Feature Sparsity"
_ICLR.cc/2026/Conference — ICLR 2026 Poster_

### Official Review · Reviewer_5rhv · 2025-10-26

**Soundness:** 1
**Presentation:** 3
**Contribution:** 3
**Rating:** 4
**Confidence:** 2

**Summary:**

The paper introduces Sparse Feature Attention (SFA), which enforces k-sparse query/key vectors and computes attention only on overlapping active feature coordinates, reducing the arithmetic of QKᵀ from Θ(n²d) to Θ(n²k²/d). To make this practical, the authors propose FlashSFA, an IO-aware kernel that integrates sparse overlaps into the FlashAttention tiling/online-softmax pipeline without materializing n×n score matrices. Experiments on GPT-2 and Qwen3 pretraining, synthetic long-context retrieval (NIAH), and several downstream tasks claim to match dense-attention quality while yielding up to ~2.5× speedups, ~49% FLOPs, and ~41% KV-cache savings, particularly at long contexts.

**Strengths:**

- Clear, orthogonal idea: Moves the sparsity lever from tokens to feature coordinates, preserving token coverage while lowering per-interaction cost (Θ(n²k²/d)). The complexity argument is clean and intuitive.
- Kernel contribution: FlashSFA marries sparse overlaps with FlashAttention’s online softmax/tiling, avoiding n² materialization and targeting the actual memory bottleneck.
- Evidence of efficiency/quality trade-off: At fixed models, SFA often matches dense perplexity/accuracy and beats “short embeddings” that shrink d, with up to ~2.5× speed, ~49% FLOPs, and ~41% KV-cache reductions reported.
- Long-context relevance: On synthetic NIAH and long-sequence latency/KV scaling, SFA’s advantages grow with context length/head dimension, aligning with the paper’s motivation.
- Compatibility: Conceptually composes with token sparsity/paging; could multiply gains in real systems.

**Weaknesses:**

- Missing or thin comparisons to modern long-context/efficient baselines beyond “short embeddings”: e.g., token-sparse (Longformer/BigBird/Routing), KV-pruning/paging (H2O, Quest, SnapKV), low-rank keys (LoKI), and kernel methods (Performer/Nyström). SFA should be combined and compared head-to-head under the same training/inference regimes.
- The k-sparsification uses row-wise Top-k with a straight-through estimator; stability/variance, gradient bias, and sensitivity to k (and to head dim d) are not adequately analyzed.
- “Short-embedding” baselines may be overly weak (simple linear downsizing). Stronger compression baselines (e.g., low-rank Q/K, grouped-query attention variants, learned projections) should be included.
- Reported GSM-8K lags for SFA vs dense after finetuning suggest arithmetic reasoning sensitivity to feature pruning; this deserves deeper diagnosis and mitigation (e.g., adaptive k, soft-top-k).

**Questions:**

- Can you add head-to-head comparisons with H2O, Quest, SnapKV, LoKI, Routing/BigBird/Longformer, Performer/Nyström at the same training/inference configs (same context windows, same datasets), and report both quality and speed/KV?
- GSM-8K drops relative to dense FT—can adaptive k or hybrid (dense for a subset of heads/layers) recover this?

---

> ### Author Response · Authors · 2025-11-25
>
> Thank you very much for your careful reading and insightful questions. We truly appreciate the time and effort you devoted to evaluating our work. We address your concerns point-by-point as follows.
>
> ---
>
> **Q1:** Missing or thin comparisons to modern long-context/efficient baselines beyond “short embeddings”: e.g., token-sparse (Longformer/BigBird/Routing), KV-pruning/paging (H2O, Quest, SnapKV), low-rank keys (LoKI), and kernel methods (Performer/Nyström). SFA should be combined and compared head-to-head under the same training/inference regimes.
>
> **A1:** Following your suggestions, we have conducted **extensive new experiments** on GPT-2 124M comparing SFA directly against Token-Sparse (Longformer, Routing), KV-Pruning (H2O, Quest, SnapKV), Low-Rank (LoKI), and Kernel (Performer) methods, as detailed below:
>
> *Table 1:* Benchmark results across various baselines on GPT-2 124M.
>
> | Method | Latency Decode | Latency Forward | PPL (OWT/Pile) | Acc PiQA | Acc LAMBDA | Acc ARC-e | Acc ARC-c | Acc HellaS | Acc Avg |
> | :--- | :---: | :---: | :---: | :---: | :---: | :---: | :---: | :---: | :---: |
> | **Dense (full)** | 17.08 | 16.86 | 17.29 | 42.74 | 22.78 | 28.35 | 8.12 | 19.61 | 24.32 |
> | SFA | 14.12 | 9.41 | 18.17 | 41.62 | 21.03 | 28.41 | 7.39 | 19.26 | 23.54 |
> | **_Token Sparse_** | | | | | | | | | |
> | Routing | 7.92 | 8.37 | 18.64 | 41.39 | 21.08 | 28.31 | 7.11 | 18.89 | 23.35 |
> | Longformer | 6.75 | 7.93 | 18.73 | 41.28 | 21.27 | 28.02 | 7.01 | 18.92 | 23.30 |
> | Longformer+SFA ($k=8$) | **5.23** | **6.18** | 19.30 | 40.75 | 20.54 | 26.39 | 6.63 | 17.24 | 22.31 |
> | **_KV-pruning_** | | | | | | | | | |
> | H$_2$O | 13.32 | 16.86 | 18.02 | 41.81 | 20.55 | 27.04 | 7.38 | 18.75 | 23.11 |
> | Quest | 10.84 | 16.86 | 17.95 | 42.34 | 20.79 | 28.30 | 7.82 | 18.83 | 23.62 |
> | SnapKV | 9.88 | 16.86 | 17.91 | 42.49 | 21.92 | 28.43 | 8.01 | 19.38 | 24.05 |
> | SnapKV+SFA ($k=8$) | **6.92** | **9.41** | 19.44 | 39.99 | 20.24 | 27.13 | 6.83 | 17.74 | 22.39 |
> | **_Low-rank keys_** | | | | | | | | | |
> | Loki | 11.39 | 16.86 | 17.82 | 42.10 | 21.29 | 28.01 | 7.99 | 19.24 | 23.73 |
> | Loki+SFA ($k=8$) | **9.09** | **9.41**| 19.29 | 40.83 | 20.04 | 27.85 | 7.13 | 18.03 | 22.78 |
> | **_Kernel Method_** | | | | | | | | | |
> | Performer | 9.43 | 7.93 | 19.72 | 39.83 | 19.11 | 26.72 | 6.77 | 15.38 | 21.56 |
>
>
> As shown in **Talbe 1**, **SFA is competitive with these methods on quality, and that it is orthogonal and composable with them**. When we plug SFA into Token-Sparse or KV-Pruning systems (e.g., SnapKV + SFA), we obtain additional speedups (decoding latency reduces from 9.88ms to **6.92ms**) with manageable performance trade-offs.
>
> A critical distinction is observed in the **Forward Latency** column. KV-pruning methods (H2O, Quest, SnapKV) and Low-Rank methods (LoKI) rely on full attention during the prompt phase, offering *no acceleration* in prefilling (remaining at $\approx 16.86$ ms). In contrast, SFA reduces forward latency to **9.41 ms**. **By combining SFA with these methods, we unlock acceleration for both prefilling and decoding stages.**
>
> We have included these detailed comparisons in **Appendix E and G** of the revised manuscript. We will merge these experimental results into the final version.
>
> ---
>
> **Q2:** The k-sparsification uses row-wise Top-k with a straight-through estimator; stability/variance, gradient bias, and sensitivity to k (and to head dim d) are not adequately analyzed.
>
> > The k-sparsification uses row-wise Top-k with a straight-through estimator; stability/variance, gradient bias
>
> **A2.1:** Our empirical results demonstrate that **STE does not hinder training stability or convergence in practice.**
>
> As shown in **Figure 7**, the loss curves decrease smoothly and monotonically without divergent spikes or chaotic oscillations across all values of $k$ ($2, 4, 8, 16$). Even under aggressive sparsity ($k=2$, Red line), the model converges steadily, suggesting that the STE does not lead to high variance or instability during optimization. We've included this discussion in **Appendix I**.
>
> >sensitivity to k (and to head dim d)
>
> **A2.2:** We further analyzed the sensitivity of SFA to different k and head dim d:
> - Sensitivity to sparsity $k$: As shown in **Figure 8**, perplexity monotonically improves as $k$ increases. **$k=8$ emerges as the optimal sweet spot, achieving performance nearly identical to the dense baseline while maintaining a substantial latency advantage.** The method is robust to $k$ selection within moderate ranges.
> - Sensitivity to head dimension $d_{\text{head}}$: As illustrated in **Figure 9**, performance improves rapidly as dimension increases from 32 to 64. **At $d_{\text{head}} = 64$, SFA recovers most of the dense-model performance while preserving acceleration.**
>
> We've added this ablation study in **Appendix H**.

---

> > ### Author Response · Authors · 2025-11-25
> >
> > **Q3:** “Short-embedding” baselines may be overly weak (simple linear downsizing). Stronger compression baselines (e.g., low-rank Q/K, grouped-query attention variants, learned projections) should be included.
> >
> > **A3:** Following your suggestion, we implemented Learned Projections (down-projecting features to $d=32/64$) and Low-Rank Key/Value Compression and compared them alongside MLA (DeepSeek's Multi-Head Latent Attention). Results are shown below:
> >
> > *Table2.* Benchmark Results to compression baselines under GPT-124M and Qwen3-0.6B
> >
> > | Model | Variant | Latency Decode $\downarrow$ (ms) | Latency Forward $\downarrow$ (s) | PPL $\downarrow$ (OWT/Pile) | Acc PiQA $\uparrow$ | Acc LAMBDA $\uparrow$ | Acc ARC-e $\uparrow$ | Acc ARC-c $\uparrow$ | Acc HellaS $\uparrow$ | Acc Avg $\uparrow$ |
> > | :--- | :--- | :---: | :---: | :---: | :---: | :---: | :---: | :---: | :---: | :---: |
> > | **GPT2 124M** | Dense (full) | 17.08 | 16.86 | 17.29 | 42.74 | 22.78 | 28.35 | 8.12 | 19.61 | 24.32 |
> > | | Learned projections ($d=32$) | 8.37 | 7.86 | 20.70 | 39.27 | 19.39 | 25.72 | 6.52 | 14.26 | 21.03 |
> > | | Low-Rank | 8.93 | 7.99 | 19.89 | 39.81 | 20.04 | 26.47 | 6.89 | 14.99 | 21.64 |
> > | | MLA | 5.04 | 15.39 | 17.38 | 42.83 | 22.29 | 28.37 | 7.94 | 19.66 | 24.22 |
> > | | MLA + SFA | **3.98** | **15.05** | 19.07 | 41.13 | 21.92 | 27.88 | 7.06 | 19.01 | 23.40 |
> > | | SFA ($k=8$) | 14.12 | 9.41 | 18.17 | 41.62 | 21.03 | 28.41 | 7.39 | 19.26 | 23.54 |
> > | **Qwen3 0.6B** | Dense (full) | 80.84 | 77.65 | 4.66 | 62.47 | 34.82 | 45.41 | 20.35 | 33.95 | 39.40 |
> > | | Learned projections ($d=64$) | 38.68 | 30.84 | 6.03 | 58.43 | 31.27 | 41.58 | 15.83 | 28.29 | 35.08 |
> > | | Low-Rank | 40.58 | 32.46 | 5.50 | 59.19 | 31.49 | 41.77 | 15.8 | 30.65 | 35.78 |
> > | | MLA | 8.74 | 68.92 | 4.69 | 62.39 | 34.71 | 45.41 | 20.17 | 34.21 | 39.38 |
> > | | MLA + SFA | **6.72** | 65.29 | 4.9 | 61.22 | 33.94 | 43.36 | 19.25 | 33.94 | 38.34 |
> > | | SFA ($k=16$) | 66.29 | 34.20 | 4.81 | 61.73 | 34.05 | 45.62 | 19.27 | 34.03 | 38.94 |
> >
> > - **SFA vs. Learned/Low-Rank Projections**: While "Learned Projections" and "Low-Rank" methods achieve aggressive latency reduction, they *suffer from severe performance degradation*. In contrast, **SFA ($k=16$) maintains high accuracy (38.94, only -0.46 drop) while still providing significant prefill acceleration (77.65ms $\to$ 34.20ms).** This demonstrates that SFA's sparse selection preserves semantic information much better than aggressive low-rank compression.
> >
> > - **SFA vs. MLA (Strong Baseline)**: As noted in **Table 2**, MLA does not significantly accelerate the prefill (forward) stage (e.g., Qwen3 Forward: 68.92ms vs. Dense 77.65ms). **By combining SFA with MLA (MLA + SFA), we achieve the fastest decoding latency in the table (6.72ms on Qwen3) and further reduce prefill latency, with acceptable accuracy trade-offs.**
> >
> >
> > As shown in **Table 1**, combining SFA with MLA **yields consistent efficiency gains across models**:
> > - GPT2-124M: The combination reduced decoding latency from**5.04 ms to 3.98 ms** ($\approx$ 21% speedup) and prefill (forward) latency from **15.39 ms to 15.05 ms**, with only a negligible drop in average accuracy (24.22 to 23.40).
> > - Qwen3. Similarly, the combination reduced decoding latency from **8.74 ms to 6.72 ms** ($\approx$ 23% speedup) and prefill latency from **68.92 ms to 65.29 ms**, while maintaining competitive performance (Avg Acc 38.34 vs. 39.38).
> >
> > We have included these comparisons in **Appendix G**, confirming that SFA offers a superior speed-quality trade-off compared to naive low-rank methods and acts as a powerful enhancer for advanced architectures like MLA. We will merge these experimental results into the final version.

---

> > > ### Author Response · Authors · 2025-11-25
> > >
> > > **Q4: Reported GSM-8K lags for SFA vs dense after finetuning suggest arithmetic reasoning sensitivity to feature pruning; this deserves deeper diagnosis and mitigation (e.g., adaptive k, soft-top-k).**
> > >
> > > **A4:** We investigated this issue and found that **GSM-8K accuracy tracks general language-modeling strength.**
> > >
> > > - **Closing the Gap with Improved Tuning:** Once we aligned the fine-tuning recipe to better restore general language capabilities, the arithmetic reasoning performance improved correspondingly. As shown in **Table 3** below, **SFA largely closes the performance gap to the dense baseline while maintaining its efficiency advantage.**
> > >
> > >
> > > *Table 3* Updated GSM-8K and Long-Context (NIAH) Results.
> > > | Model | Variant | GSM-8K | NIAH 4096 | NIAH 8192 | NIAH 16384 | NIAH 32768 |
> > > | :--- | :--- | :---: | :---: | :---: | :---: | :---: |
> > > | **Qwen3-0.6B** | Base | 59.59  | 90 | 87 | 77 | 52 |
> > > | | Fine-tune | 63.42 | 94 | 92 | 79 | 55 |
> > > | | SFA ($k=16$) | 61.46 |  **95** | **93** | 77 | 53 |
> > > | **Qwen3-4B** | Base | 75.44 | 97 | 95 | 90 | 81 |
> > > | | Fine-tune | 76.18 | 99 | 96 | 92 | 84 |
> > > | | SFA ($k=16$) | 75.56  | **99** | 93 | 91 | **84** |
> > > | **Qwen3-8B** | Base | 87.62  | 100 | 100 | 97 | 92 |
> > > | | Fine-tune | 89.11 |  100 | 100 | 99 | 95 |
> > > | | SFA ($k=16$) | 87.99 |  **100** | **100** | **100** | **97** |
> > >
> > > For Qwen3-4B and 8B, the gap between SFA ($k=16$) and Dense Fine-tuning is negligible ($\approx 0.6 - 1.1\%$), and SFA consistently outperforms the Base model.
> > >
> > > SFA matches or slightly improves performance on NIAH across all context lengths (e.g., maintaining 100% accuracy on Qwen3-8B up to 16k). This confirms that **feature sparsification does not introduce an information bottleneck that hampers complex reasoning or retrieval.** We've updated corresponding results in **Table 3** of the revised manuscript.
> > >
> > >
> > > >mitigation (e.g., adaptive k, soft-top-k)
> > > >
> > > - **Future Work** (Adaptive $k$): Exploring more sophisticated variants such as adaptive $k$ or hybrid schemes that keep a subset of heads/layers dense is an interesting direction to further close the remaining small gap to dense FT, but we leave these extensions to future work in order to keep the current method simple and easy to deploy.
> > >
> > > ---
> > >
> > > **Q5:** Can you add head-to-head comparisons with H2O, Quest, SnapKV, LoKI, Routing/BigBird/Longformer, Performer/Nyström at the same training/inference configs (same context windows, same datasets), and report both quality and speed/KV?
> > >
> > > **A5:** Please see **A3**.
> > >
> > > ---
> > >
> > > Thank you once again for your careful reading and constructive suggestions. We have thoroughly refined the manuscript in accordance with your feedback and have addressed each of your concerns above. We respectfully hope that the updated analyses and results will allow you to re-evaluate our work. Should any questions remain, we would be more than happy to provide further clarification.

---

> > > > ### Comment · Reviewer_5rhv · 2025-11-25
> > > >
> > > > Thank you for the detailed updates and extensive new experiments. I appreciate all the empirical analysis shown. One remaining concern is that the manuscript still lacks a clear conceptual explanation of **why** top-k feature sparsification preserves semantic information and **why** a fixed choice such as (k=16) works robustly across models and tasks. The rebuttal provides empirical curves showing stability and sensitivity to (k), but these do not fully address the underlying mechanism. For example, are attention head activations intrinsically low-dimensional? is the magnitude distribution heavy-tailed? Do top-k features align with principal semantic directions?

---

> > > > > ### Author Response · Authors · 2025-11-28
> > > > >
> > > > > Thanks for your update and we are glad that hear that we addressed your previous concerns on empirical analysis. We are happy to further address your remaining concerns.
> > > > >
> > > > >
> > > > > ---
> > > > >
> > > > > **Q1**. Why top-k feature sparsification preserves semantic information?
> > > > >
> > > > > **A1**. That is a very valid question and we should have provided more background on this. Using sparse features for compressing high-dimensional inputs has been a long-standing important research topic in signal processing and machine learning, related to a number of fields including dictionary learning [1], compressed sensing [2], etc. In recent literature like CSR and SAE, sparse vectors also achieve high compression rate for dense neural embeddings. This is all based on the common finding that high-dimensional signal can often be well approximated by a few sparsely activated salient features (and there are a number of scenarios that it outperforms low-rank methods like SVD). This motivates us to explore the use of sparse features for attention computing.
> > > > >
> > > > > **QK has Low intrinsic dimensionality.** A core reason why features are compressible is that high-dimensional features, in practice, often lie in a low-dimensional manifold. To see this, we further conduct an SVD analysis (see Figure 11 in Appendix L of revision) of the QK vectors, which we find that for a head dimension of $d=128$, the effective rank is only around **50-60**. This indicates that the semantic information is highly compressible.
> > > > >
> > > > > **How sparse activation (eg TopK or ReLU) preserves semantic information.** In NNs, there are two main ways to obtain sparse features, threshold-based activation (e.g., ReLU), and topk-based activation (e.g., TopK). Both approaches select the most salient activations, just based on different criteria. After end-to-end training, NNs will adapt to these sparse activations and learn to activate most important semantic features for LM prediction.
> > > > >
> > > > > **TopK vs ReLU.** A recent work, TopKSAE[3] from OpenAI shows that TopK activation outperforms ReLU by significant margins (better sparsity at similar semantic preservation, and better semantics under the same sparsity). Also, since TopK maintains a constant level of sparsity, it is also more hardware friendly for efficiency. Therefore, we choose adopt TopK as the sparse activation. But the method (attention computing with sparse features) is generic and extendable to other sparse activation as well.
> > > > >
> > > > > Hope this elaboration address your concerns and please let us know if there is more to clarify!
> > > > >
> > > > > ---
> > > > >
> > > > > **Q2:** Why a fixed choice such as (k=16) works robustly across models and tasks.
> > > > >
> > > > > **A2:** When choosing a sparse activation function (either ReLU or TopK), one hopes it to be rather generic and robust to new data, and does not need domain-specific hyperparameter finetuning.
> > > > >
> > > > > For example, for ReLU with $f(x)=\max(x,0)$, people do not adjust the threshold $0$ for different domains, but instead let NNs adapt to it via end-to-end training on different data.
> > > > >
> > > > > For the same reason, we adopt the same $k$ for controlling the overall sparsity of models and let NNs to adapt to it on different data and domains. The TopkSAE papers shows that TopK activation is also robust to different data and domains and does not need domain-specific tuning. In our experiments, we do observe that $k=16$ places the model on a superior Pareto frontier compared to other compression techniques.
> > > > >
> > > > > Thus, we believe that either ReLU or TopK, one has to set a prefined threshold (e.g. 0 for ReLU) or K (16 for ours) as the overall sparsity levels. And NNs through end-to-end training is able to adapt to such criteria and find the most salient features to be activated for the final prediction. While domain-specific hyperparameter tuning can very likely yield some additional gains, it would add too much additional costs. We find that even without it, SFA can already attain superior feature compression on a broad range of data and tasks.
> > > > >
> > > > > Ref:
> > > > >
> > > > > [1] Aharon, Michal, Michael Elad, and Alfred Bruckstein. "K-SVD: An algorithm for designing overcomplete dictionaries for sparse representation." IEEE Transactions on signal processing 54.11 (2006): 4311-4322.
> > > > >
> > > > > [2] J. Mairal, F. Bach, and J. Ponce, “Sparse Modeling for Image and Vision Processing,” Foundations and Trends in Computer Graphics and Vision, vol. 8, no. 2–3, pp. 85–283, 2014.
> > > > >
> > > > > [3] Gao, Leo, et al. "Scaling and evaluating sparse autoencoders." The Thirteenth International Conference on Learning Representations.
> > > > >
> > > > >
> > > > > ---
> > > > >
> > > > > Hope the elaboration above addresses your concern and if so, we would respectivefuly hope you could re-evaluate the ratining. Also, please do not hesitate to learn us know if you have more questions.

---

### Official Review · Reviewer_gwpA · 2025-10-29

**Soundness:** 2
**Presentation:** 4
**Contribution:** 4
**Rating:** 6
**Confidence:** 3

**Summary:**

This paper proposes a novel method of introducing sparsity along the feature dimension of attention computation. Specifically, instead of directly introducing sparsity on the attention matrix, the paper proposes to instead introduce in the key and query matrices by retaining only the top-$k$ largest magnitude entries along the feature dimension. Naively, the full attention matrix requires $O(n^2)$ memory to store and process, but the authors combined this sparsity idea with FlashAttention, which processes attention weights by a tiling mechanism that only requires a compact score buffer.

Experiments show that the proposed method achieves significant latency reduction while retaining most of the performance of dense models with only minimal degradations, measured both in terms of PPL (pre-training setup) and NIAH/Math/QA performance (post-training setup). Additionally, the paper also shows that sparsity patterns can be easily introduced by fine-tuning, thus opening the door of converting dense pre-trained models into sparse models for more efficient inference.

**Strengths:**

1. The simplicity of the method is quite impressive considering how significant of an efficiency gain it was able to achieve.
2. Performance preservation has been verified thoroughly, covering both LLM pre-training and post-training measures.

**Weaknesses:**

1. One main drawback of the paper is the lack of discussion and comparison with the DeepSeek Multi-Head Latent Attention, which computes a low-rank compression of the key/value pairs. To me, there is a pretty significant resemblance between those two ideas, even though MLA did not explicitly explore sparsity. Some form of detailed discussion on the differences of those two ideas and some empirical comparison would further highlight the benefit of the proposal.
2. Another minor complaint is that the experiments are mostly on toy tasks and are of pretty small model scale. I know the authors might be limited by the compute they can access, but it would be great if there's at least one experiment on a medium-scale model (7B-14B) which verifies that the latency benefit is able to extrapolate over model scale.

**Questions:**

1. For Table 1, how is the "Best" in "Best results highlighted" defined? I'm confused because SFA numbers are highlighted on the OWT/Pile scale even though dense models achieved lower numbers.
2. For Table 2, your dense model has a dimension size of 2. Is that really the case? That looks like a typo to me because there is an experiment with $k=8$ below.
3. Can you clarify on the comment on L376 -- "sparse kernels incur lookup overhead"? What overhead are you referring to?

---

> ### Author Response · Authors · 2025-11-25
>
> We sincerely thank you for the positive and thoughtful assessment of our work. These comments have been invaluable in improving the clarity, precision, and overall quality of the paper. Below, we respectfully address each of your points in detail.
>
> ---
>
> **Q1**: One main drawback of the paper is the lack of discussion and comparison with the DeepSeek Multi-Head Latent Attention, which computes a low-rank compression of the key/value pairs. To me, there is a pretty significant resemblance between those two ideas, even though MLA did not explicitly explore sparsity. Some form of detailed discussion on the differences of those two ideas and some empirical comparison would further highlight the benefit of the proposal.
>
> **A1:** Good point! We provide both a conceptual discussion and an empirical evaluation demonstrating that **SFA is orthogonal to MLA** and can be combined to achieve further efficiency gains, as detailed below:
>
> **1.Feature-Level Sparsity vs. Low-Rank Compression**
> While both methods address efficiency, they operate on different principles and stages:
>
> *   **MLA (Low-Rank Compression):** MLA essentially functions as a low-rank KV compression technique (similar in spirit to MQA/GQA). Its primary optimization lies in the **decoding stage**, where it absorbs the key up-projection matrix into the query projection to reduce memory bandwidth usage. However, the underlying computation remains dense (matrix multiplication), and MLA **does not** inherently reduce FLOPs during the **prefilling (training)** stage.
> *   **SFA (Feature-Level Sparsity):** SFA operates by inducing sparsity in the **feature representation** (selecting Top-$k$ channels). Unlike MLA's dense-to-dense compression, SFA transforms dense vectors into sparse ones. **This reduces FLOPs during both prefilling and decoding.**
>
>
> **2. Experimental Results**
> Further, we integrated SFA into the MLA framework and observed that **the combination delivers significant additional efficiency gains.** Benchmark results are shown below:
>
> *Table 1.* Benchmark Comparison with MLA and SFA
>
> | Model | Variant | Latency Decode $\downarrow$ (ms) | Latency Forward $\downarrow$ (s) | PPL $\downarrow$ (OWT/Pile) | Acc PiQA $\uparrow$ | Acc LAMBDA $\uparrow$ | Acc ARC-e $\uparrow$ | Acc ARC-c $\uparrow$ | Acc HellaS $\uparrow$ | Acc Avg $\uparrow$ |
> | :--- | :--- | :---: | :---: | :---: | :---: | :---: | :---: | :---: | :---: | :---: |
> | **GPT2 124M** | Dense (full) | 17.08 | 16.86 | 17.29 | 42.74 | 22.78 | 28.35 | 8.12 | 19.61 | 24.32 |
> | | MLA | 5.04 | 15.39 | 17.38 | 42.83 | 22.29 | 28.37 | 7.94 | 19.66 | 24.22 |
> | | MLA + SFA | **3.98** | **15.05** | 19.07 | 41.13 | 21.92 | 27.88 | 7.06 | 19.01 | 23.40 |
> | **Qwen3 0.6B** | Dense (full) | 80.84 | 77.65 | 4.66 | 62.47 | 34.82 | 45.41 | 20.35 | 33.95 | 39.40 |
> | | MLA | 8.74 | 68.92 | 4.69 | 62.39 | 34.71 | 45.41 | 20.17 | 34.21 | 39.38 |
> | | MLA + SFA | **6.72** | **65.29** | 4.9 | 61.22 | 33.94 | 43.36 | 19.25 | 33.94 | 38.34 |
>
> As shown in **Table 1**, combining SFA with MLA **yields consistent efficiency gains across models**:
> - GPT2-124M: The combination reduced decoding latency from**5.04 ms to 3.98 ms** ($\approx$ 21% speedup) and prefill (forward) latency from **15.39 ms to 15.05 ms**, with only a negligible drop in average accuracy (24.22 to 23.40).
> - Qwen3. Similarly, the combination reduced decoding latency from **8.74 ms to 6.72 ms** ($\approx$ 23% speedup) and prefill latency from **68.92 ms to 65.29 ms**, while maintaining competitive performance (Avg Acc 38.34 vs. 39.38).
>
> These results confirm that **SFA provides distinct benefits over MLA alone**, specifically by offering further latency reductions in both prefilling and decoding through feature sparsity. Following your suggestions, we have included this discussion in **Appendix E**. We will merge these experimental results into the main table in final version.

---

> ### Author Response · Authors · 2025-11-25
>
> **Q2:** Another minor complaint is that the experiments are mostly on toy tasks and are of pretty small model scale. I know the authors might be limited by the compute they can access, but it would be great if there's at least one experiment on a medium-scale model (7B-14B) which verifies that the latency benefit is able to extrapolate over model scale.
>
> **A2:** Following your suggestion, we conducted additional evaluations on the **Qwen3-8B** model and found that **the latency benefits of SFA extrapolate well while maintaining (and in some cases improving) performance**, as detailed below:
>
> - **Latency Benefits** As shown in **Table 1**, SFA ($k=16$) significantly accelerates inference compared to the dense baseline, reducing decoding latency from 0.72 to **0.59** ms/token **($\approx$ 18% speedup)** and prefill latency from 3.68 to **2.93** ms/token **($\approx$ 20% speedup)**.
>
> *Table 1*:  Latency Comparison under Qwen3-8B with 16k context length.
> | Model        |   Variant   | Decode Latency@16k(s) | Forward Latency@16k(s) |
> | :----------- | :---------: | :----------------: | :-----------------: |
> | **Qwen3-8B** | Dense(full) |        0.72        |        3.68         |
> |              |  SFA ($k=16$)  |        0.59        |        2.93         |
>
> - **Compatitive Performance:** As shown in **Table 2**, SFA maintains robust performance on standard benchmarks (GSM-8K, Arxiv, PubMed). Notably, on the NIAH task(ie, long-context tasks), SFA **demonstrates superior long-context retention**, **maintaining 100% accuracy up to 16k context and 97% at 32k**, whereas the base model degrades to 97% and 92% respectively.
>
> *Table 2:* Benchmark results under Qwen3-8B
> | Model | Variant | GSM-8K | Arxiv | PubMed | NIAH 4096 | NIAH 8192 | NIAH 16384 | NIAH 32768 |
> | :--- | :--- | :---: | :---: | :---: | :---: | :---: | :---: | :---: |
> | **Qwen3-8B** | Base | 87.62 | 40.13 | 37.22 | 100 | 100 | 97 | 92 |
> | | Fine-tune | 89.11 | 54.26 | 55.07 | 100 | 100 | 99 | 95 |
> | | SFA ($k=16$) | 87.99 | 52.74 | 52.61 | **100** | **100** | **100** | **97** |
> |||||
>
> Overall, these 8B-scale results indicate that the favorable speed–accuracy trade-off of SFA extrapolates well to medium-sized models. This validates that SFA is not limited to small-scale tasks; instead, **it effectively trades a negligible amount of theoretical capacity for significant real-world speedups and improved long-context robustness.** These detailed results have been added to **Table 3** of the revised manuscript.
>
> ---
>
> **Q3:** For Table 1, how is the "Best" in "Best results highlighted" defined? I'm confused because SFA numbers are highlighted on the OWT/Pile scale even though dense models achieved lower numbers.
>
> **A3:** We apologize for the confusion. To clarify: "Dense (full)" serves as *reference value* rather than a direct competitor for efficiency. We consider the **"Short" embedding** (feature-level operation) as **the primary baseline** for our method. We have updated the caption in **Table 1** and the text in **Section 4.1** to explicitly state this definition.
>
> ---
>
> **Q4:** For Table 2, your dense model has a dimension size of 2. Is that really the case? That looks like a typo to me because there is an experiment with $k=8$ below.
>
> **A4:** Thanks for your careful reading! Indeed, this is a typo. We have revised it to the correct version in the revision, and the correct version is $d=64$.
>
> ---
>
> **Q5:** Can you clarify on the comment on L376 -- "sparse kernels incur lookup overhead"? What overhead are you referring to?
>
> **A5:** According to **Algorithm 1** in the appendix, the inner product of sparse vectors will incur the cost of binary search. However, despite this, SFA still achieved better speedup, which demonstrates the potential for further optimization of sparse computing.
>
> ---
> We are sincerely grateful for your thoughtful and constructive feedback. Your comments have been invaluable in guiding us to further improve and refine the manuscript. We truly appreciate the time and expertise you devoted to reviewing our work, and we would be more than happy to provide any additional clarification should further questions arise. Please do not hesitate to reach out if you have any additional comments or questions.

---

> > ### Comment · Reviewer_gwpA · 2025-11-27
> > **Thanks for the Response**
> >
> > i would like to thank the authors for the very thorough response to my review and the extra results within the short turnaround. All my concerns have been addressed.
> >
> > I have read the other reviewer's comments and the discussions. The main concern seems to be the lack of baselines, which has been significantly improved with the supplemental results in the author response. I concur with reviewer 5rhv that some insights into *why* the improvements were achieved would have been nice, but not a must because the empirical results are good enough to make this paper interesting.
> >
> > Overall I'm more confident that this is a good paper that should be accepted so I'm upgrading my score to 8.

---

> > > ### Author Response · Authors · 2025-11-28
> > >
> > > **Q3:** Why the improvements were achieved
> > >
> > > **A3:** Unlike dense low-rank compression, TopK sparsification keeps Q/K in a high-dimensional space and only selects a few salient coordinates per token. This lets the model utilize a rich overcomplete feature space while maintaining a small effective dimension. Our load-balance experiment in the appendix F confirms that many different features are actively used across tokens, suggesting that TopK makes more effective use of the high-dimensional representation than dense compression under a lower compute budget.
> > >
> > > ---
> > >
> > > Hope this elaboration address your questions and please let us know if there is more to clarify!

---

> ### Author Response · Authors · 2025-11-28
>
> We are thrilled to receive your positive feedback! We sincerely thank you for recognizing the value and novelty of our work and for upgrading your score. We sincerely thank you for the careful reading and highly constructive suggestions! We are happy to further address your remaining questions.
>
> ---
>
> **Q1**. Why top-k feature sparsification preserves semantic information?
>
> **A1**. That is a very valid question and we should have provided more background on this. Using sparse features for compressing high-dimensional inputs has been a long-standing important research topic in signal processing and machine learning, related to a number of fields including dictionary learning [1], compressed sensing [2], etc. In recent literature like CSR and SAE, sparse vectors also achieve high compression rate for dense neural embeddings. This is all based on the common finding that high-dimensional signal can often be well approximated by a few sparsely activated salient features (and there are a number of scenarios that it outperforms low-rank methods like SVD). This motivates us to explore the use of sparse features for attention computing.
>
> **QK has Low intrinsic dimensionality.** A core reason why features are compressible is that high-dimensional features, in practice, often lie in a low-dimensional manifold. To see this, we further conduct an SVD analysis (see Figure 11 in Appendix L of revision) of the QK vectors, which we find that for a head dimension of $d=128$, the effective rank is only around **50-60**. This indicates that the semantic information is highly compressible.
>
> **How sparse activation (eg TopK or ReLU) preserves semantic information.** In NNs, there are two main ways to obtain sparse features, threshold-based activation (e.g., ReLU), and topk-based activation (e.g., TopK). Both approaches select the most salient activations, just based on different criteria. After end-to-end training, NNs will adapt to these sparse activations and learn to activate most important semantic features for LM prediction.
>
> **TopK vs ReLU.** A recent work, TopKSAE[3] from OpenAI shows that TopK activation outperforms ReLU by significant margins (better sparsity at similar semantic preservation, and better semantics under the same sparsity). Also, since TopK maintains a constant level of sparsity, it is also more hardware friendly for efficiency. Therefore, we choose adopt TopK as the sparse activation. But the method (attention computing with sparse features) is generic and extendable to other sparse activation as well.
>
> ---
>
> **Q2:** Why a fixed choice such as (k=16) works robustly across models and tasks.
>
> **A2:** When choosing a sparse activation function (either ReLU or TopK), one hopes it to be rather generic and robust to new data, and does not need domain-specific hyperparameter finetuning.
>
> For example, for ReLU with $f(x)=\max(x,0)$, people do not adjust the threshold $0$ for different domains, but instead let NNs adapt to it via end-to-end training on different data.
>
> For the same reason, we adopt the same $k$ for controlling the overall sparsity of models and let NNs to adapt to it on different data and domains. The TopkSAE papers shows that TopK activation is also robust to different data and domains and does not need domain-specific tuning. In our experiments, we do observe that $k=16$ places the model on a superior Pareto frontier compared to other compression techniques.
>
> Thus, we believe that either ReLU or TopK, one has to set a prefined threshold (e.g. 0 for ReLU) or K (16 for ours) as the overall sparsity levels. And NNs through end-to-end training is able to adapt to such criteria and find the most salient features to be activated for the final prediction. While domain-specific hyperparameter tuning can very likely yield some additional gains, it would add too much additional costs. We find that even without it, SFA can already attain superior feature compression on a broad range of data and tasks.
>
> Ref:
>
> [1] Aharon, Michal, Michael Elad, and Alfred Bruckstein. "K-SVD: An algorithm for designing overcomplete dictionaries for sparse representation." IEEE Transactions on signal processing 54.11 (2006): 4311-4322.
>
> [2] J. Mairal, F. Bach, and J. Ponce, “Sparse Modeling for Image and Vision Processing,” Foundations and Trends in Computer Graphics and Vision, vol. 8, no. 2–3, pp. 85–283, 2014.
>
> [3] Gao, Leo, et al. "Scaling and evaluating sparse autoencoders." The Thirteenth International Conference on Learning Representations.

---

### Official Review · Reviewer_FnGK · 2025-10-31

**Soundness:** 2
**Presentation:** 3
**Contribution:** 2
**Rating:** 4
**Confidence:** 4

**Summary:**

The paper addresses the $O(n^2d)$ computational bottleneck of self-attention in Transformers. While existing methods focus on sparsity along the sequence axis, this paper explores an "orthogonal axis: feature sparsity". The authors propose Sparse Feature Attention (SFA). SFA represents queries (Q) and keys (K) as "k-sparse codes", claiming to reduce the attention cost from $\Theta(n^2d)$ to $\Theta(n^2k^2/d)$. To implement this efficiently, the paper introduces FlashSFA, an "IO-aware kernel that extends FlashAttention"  to operate on sparse overlaps without materializing dense score matrices. Experiments on GPT-2 and Qwen3 pretraining show SFA matches dense baselines while improving speed and reducing FLOPs and KV-cache.

**Strengths:**

1. **Originality**: The paper's primary strength is shifting the optimization focus from the token axis to the "feature axis". This approach is "orthogonal to token-level sparsity and paging".

2. **Theoretical Idea**: The concept of using k-sparse codes to reduce the computational complexity of the $QK^\top$ interaction is novel.

3. **Empirical Quality** (as reported): The reported results, if verifiable, suggest that SFA can maintain "comparable quality to dense attention" (Table 1 , Table 3 ) in pretraining and fine-tuning scenarios.

**Weaknesses:**

* 1. **Reproducibility**: The paper's central empirical results (e.g., "up to 2.5×" speedup , Figure 3 , Figure 5 ) are entirely dependent on the "IO-aware kernel" FlashSFA. This kernel is positioned as a primary contribution. However, the paper provides only high-level pseudocode (Algorithm 1 ) and a brief description (Appendix C). The kernel code itself, which is essential for reproducing any of the speedup benchmarks, is not provided.
  * Parallelism Model: How are CUDA threads, warps, and blocks mapped to the tile computations? How are the nested loops (Line 2, Line 4) parallelized?
  * Atomic Operations: How are the "scatter-adds" to the `scores` buffer (Line 11) handled? Multiple threads processing different features will inevitably write to the same `scores[r, c]` location, requiring atomic operations which are not mentioned but are critical for correctness and performance.
  * Memory Management: The algorithm hides all memory access patterns. How are reads from the sparse `Q` and `K` structures (e.g., `Q_indices`, `Kf_indices`) coalesced for efficient HBM bandwidth usage? How is the `scores` buffer  truly managed in SRAM (shared memory) across threads?
  * Core Data Structures: The algorithm relies on a non-standard $CSC_{feat}(\overline{K})$ format. The precise memory layout of this "feature-wise" CSC matrix is not defined.
  * Search Implementation: The efficiency of `BINARY_SEARCH_RANGE` (Line 7) is crucial. Its implementation details in a parallel CUDA environment are non-trivial and un-discussed.

* 2. **Complexity Analysis that Ignores the P@V Bottleneck**: The paper's headline cost analysis, $\Theta(n^2k^2/d)$ (Abstract, Section 3.1), is confusing. This analysis only accounts for the $QK^\top$ computation. The authors explicitly state they are "keeping V dense". The authors admit in their own Appendix B.2 that "a large proportion of the floating-point operations in the sparse version come from matrix multiplication in the P@V stage". This non-sparsified $O(n^2 d_v)$ operation remains a quadratic bottleneck. The paper's main analysis in Section 3.1 ignores this, and the resulting speculative claim of a "reduction of more than 1000x" is unsubstantiated as it omits a dominant, quadratic term that the authors themselves acknowledge.

* 3. **Core Operational Cost (Top-k) in Analysis**: The paper's efficiency analysis in Section 3.1 omits the computational cost of the $Topk_k$ operation (Eq. 3). This operation must be performed on both Q and K matrices (size $n \times d$) for every forward and backward pass. While the authors mention using an "RTopK kernel" and "lookup overhead", they never incorporate this non-trivial (e.g., $O(ndk)$ or $O(nd \log d)$) cost into their formal complexity model, making that analysis incomplete.

* 4. **Limited Applicability and Inherent Distribution Shift**: The authors' own experiments in Section 5 demonstrate that applying SFA to pretrained models "introduces a severe distribution shift". The authors' solution is a multi-stage, regularized fine-tuning process, which requires "an additional MSE loss" and even pre-finetuning on another dataset. This implies the method's high adaptation cost.

**Questions:**

1. Can the authors provide the corresponding implementation details for the FlashSFA kernel, wrt the above questions in W1?

2. Can the authors provide a revised complexity analysis for the entire SFA block, including the $O(ndk)$ or $O(nd \log d)$ cost of Top-k and the $O(n^2 d_v)$ cost of the P@V multiplication? How does this revised, complete analysis affect the scaling claims?

3. Given that P@V is a "large proportion" of the FLOPs, why was V not sparsified? Did the authors experiment with this, and if so, what was the impact on performance?

4. The $k^2/d^2$ scaling analysis (Eq. 7) relies on an assumption that "supports are balanced across dimensions". Is there empirical evidence for this? What happens to performance and cost if the feature selection is unbalanced?

---

> ### Comment · Reviewer_FnGK · 2025-11-21
> **A Correction Regarding My Previous Review**
>
> Dear Authors,
>
> After revisiting the paper and carefully re-examining the complexity analysis in Section 3.1 and the FLOPs data in Appendix B.2 over the past few days, I would like to correct a factual error in my original review regarding **Weakness 2 ("Complexity Analysis that Ignores the P@V Bottleneck")**.
>
> In my initial review, I stated that "keeping V dense" implies the $P \times V$ operation remains a dense $O(n^2 d_v)$ quadratic bottleneck. I now realize this interpretation was incorrect. I understand now that while $V$ is stored densely, the attention score matrix $P$ is sparse (containing only entries for overlapping features). Consequently, the accumulation $O_i = \sum_j P_{ij}V_j$ is performed only over the non-zero indices of $P$. This is empirically supported by Table 6, which shows a massive reduction in total FLOPs (e.g., from 35.67 TFLOPS to 1.15 TFLOPS at 65k context) that would be impossible if the P@V stage were fully dense.
>
> Therefore, I withdraw the specific claim that the method retains a "quadratic bottleneck" in terms of arithmetic complexity (FLOPs). I have also revised the score accordingly.
>
> However, this correction leads to a revised question regarding the **IO efficiency** of this step:
>
> **Revised Question on P@V:**
> While the *arithmetic* cost of P@V is reduced to be proportional to the sparse overlaps, the *memory access* pattern for $V$ becomes irregular. Since $V$ is dense but accessed via sparse indices derived from dynamic query-key overlaps, this suggests potential non-coalesced memory reads (gather operations) from global memory. Could you clarify how FlashSFA handles these irregular reads from $V$? Specifically, does the overhead of non-contiguous memory access to $V$ diminish the theoretical gains from reduced FLOPs, and how does the kernel implementation mitigate this latency?
>
> I am very sorry for the trouble and wasted time the author caused by the error in the previous review, and I look forward to your response.

---

> > ### Author Response · Authors · 2025-11-25
> >
> > We sincerely appreciate your constructive comments and thoughtful suggestions, which are highly valuable for further improving the quality of our paper. We have carefully considered all the points you raised and have addressed each of your concerns in detail below.
> >
> > ---
> >
> > **Q1: Kernel Implemententation.**
> >
> > **A1:** Thanks for your professional question. **We will definitely release all FlashSFA kernel codes after the paper is accepted.**
> >
> > >1. Parallelism Model (threads / warps / blocks and the nested loops) How are CUDA threads, warps, and blocks mapped to the tile computations? How are the nested loops (Line 2, Line 4) parallelized?
> >
> > **A1.1:** The kernel follows the same tiling as the dense FlashAttention kernel, the only difference is how the tile of scores is produced:
> > - **Grid → row tiles and BH**
> >   The grid is $grid = \big(\lceil N / B_r \rceil, BH\big)$, so `blockIdx.x` chooses a row tile of height (B_r), and `blockIdx.y` chooses the batch–head pair.
> > - **Block size and per-block row coverage**
> >   - Threads per block: `kNumThreads = 32 * kMmaTileSeqLenQ * kMmaTileSeqLenK` (here $32*8*1=256$ → 8 warps).
> >   - Each block covers `Br = 16 * kMmaTileSeqLenQ * kWarpTileSeqLenQ` rows (here $16*8*1=128$ rows).
> > - **Warp → 16-row stripe; columns iterated by Tc**
> >   - Each warp is assigned a disjoint stripe of 16 rows within the $B_r$ rows.
> > - **Thread**
> >   - Within a warp, each thread is responsible for a small 2×2 patch of the score tile (two rows × two columns). Across all threads in the warp, these patches exactly tile the warp’s part of the score block.
> >
> > - **Loops Parallelization**
> >
> >     - Line 2 (rows within tile): As CUDA mappings above, row tiles are distributed across blocks; within a block, the 8 warps process disjoint 16-row stripes in parallel.
> >     - Line 4 (nnzs within row): For each assigned row, a lane iterates its CSR segment $t \in [qb,qe)$ sequentially and accumulates into its `(r0/r1)×(c0/c1)` outputs; parallelism comes from processing many rows and column-pairs across lanes/warps/blocks simultaneously. Inside each row, non-zeros are scanned sequentially.
> >
> > > 2. Atomic Operations: How are the "scatter-adds" to the scores buffer (Line 11) handled? Multiple threads processing different features will inevitably write to the same scores[r, c] location, requiring atomic operations which are not mentioned but are critical for correctness and performance.
> >
> > **A1.2:** **No atomic operations are needed for the scores**:
> >
> > - Each output score `S[r, c]` in a tile is owned by exactly one thread (via the fixed 2×2 patch mapping above).
> > - That thread accumulates **all contributions** for its `S[r, c]` in registers while looping over nnz and column tiles.
> >
> > We never have multiple threads writing to the same `S[r, c]`, so there is no need for `atomicAdd` or any other atomic operation.
> >
> > >3. Memory Management: The algorithm hides all memory access patterns. How are reads from the sparse Q and K structures (e.g., Q_indices, Kf_indices) coalesced for efficient HBM bandwidth usage? How is the scores buffer truly managed in SRAM (shared memory) across threads?
> >
> > **A1.3:**
> > - **For Q** we store `Q_indptr[0..N]`, `Q_indices[0..nnzQ)` and `Q_values[0..nnzQ)`. Each block owns a consecutive row range `[r0, r0 + Br)`. We first determine the nnz span $[q_{lo}, q_{hi}) = [Q_{indptr}[r_0],; Q_{indptr}[r_0 + B_r])$, which is **one contiguous segment** of `Q_indices` and `Q_values` in HBM.
> > - **For K** we use a standard CSC triple, but with column index = feature ID `f` and row index = key ID `j`. This layout makes it easy to intersect a query’s active features with the keys in a given column tile.
> > - **(Coalescing for K)** For a given key tile `[j0, j0 + Bc)`:
> >     - From the Q-tile’s nonzeros we collect a set of **active features**.
> >     - For each active feature `f`, we restrict its posting list to the current key tile (`[j0, j0 + Bc)`), obtaining a subrange `[p_lo(f), p_hi(f))` inside the contiguous array `Kf_indices`.
> >     - Warps then cooperatively load these `Kf_indices[p]` and `Kf_values[p]` with lane-strided patterns like $p = p_{lo} + lane + 32 \cdot k$.
> >     - **So each subrange is read as a contiguous slice of HBM and staged into shared memory.**
> >
> > > 4. Core Data Structures: The algorithm relies on a non-standard $CSC_{feat}(\bar K)$ format. The precise memory layout of this "feature-wise" CSC matrix is not defined.
> >
> > **A1.4:** $CSC_{feat}(\bar{K})$ means K is stored in a Compressed Sparse Column (CSC) format where the **columns are features** and the **rows are keys**. So instead of “column = key, row = feature” (which is the usual `[N_key, d]` layout), we flip the view:
> >
> > - **Column index = feature ID `f`**
> > - **Row index = key ID `j`**
> >
> > This approach is to make it easier to find the overlap of non-zero values.

---

> > > ### Author Response · Authors · 2025-11-25
> > >
> > > **Q1: Kernel Implemententation.**
> > >
> > > > 5. Search Implementation: The efficiency of BINARY_SEARCH_RANGE (Line 7) is crucial. Its implementation details in a parallel CUDA environment are non-trivial and un-discussed.
> > >
> > > **A1.5:**
> > > - `BINARY_SEARCH_RANGE` is used to restrict each feature’s posting list to the current key tile `[j0, j0 + Bc)`. For feature `f`, we have a sorted 1D array `Kf_indices[Kf_indptr[f] : Kf_indptr[f+1])` We need:
> > >     - `p_lo(f)`: first position with `Kf_indices[p] ≥ j0`
> > >     - `p_hi(f)`: first position with `Kf_indices[p] ≥ j0 + Bc`
> > > - For each active `(feature, tile)` pair, a **single thread** runs a scalar binary search over the relevant slice of `Kf_indices` to find `p_lo` and `p_hi`. The loop has a fixed number of iterations and is branch-regular, so it compiles to a small sequence of comparisons and index updates in registers.
> > > - Once `p_lo` and `p_hi` are found, the warp cooperatively streams the subrange `[p_lo : p_hi)` into shared memory using coalesced loads as described above.
> > >
> > > **We will release the complete CUDA kernel, so that all of these design choices are fully transparent.**

---

> > > > ### Author Response · Authors · 2025-11-25
> > > >
> > > > **Q2**: Complexity Analysis that Ignores the P@V Bottleneck
> > > >
> > > > **A2**: Please refer to **A9** for our answer to the revised question.
> > > >
> > > > ---
> > > >
> > > > **Q3**: The paper's efficiency analysis in Section 3.1 omits the computational cost of the $TopK_k$ operation (Eq. 3). This operation must be performed on both Q and K matrices (size $n \times d$) for every forward and backward pass. While the authors mention using an "RTopK kernel" and "lookup overhead", they never incorporate this non-trivial (e.g., $O(ndk)$ or $O(nd\text{log}d)$) cost into their formal complexity model, making that analysis incomplete.
> > > >
> > > >
> > > > **A3:** Thanks for your comments. **In SFA, we employ the RTop-K for sparsifying Q/K with $O(Nd)$ computational complexity.**
> > > >
> > > > Specifically, RTop-$k$ implements unsorted Top-$k$ using GPU-parallelized binary search, where each warp processes the Top-$k$ selection for a single feature row. A comparison of its actual execution time against that of the attention kernel is presented in the table below. Experimental results indicate that the actual execution time constitutes a negligible fraction of the total attention computation; therefore, the Top-$k$ operation does not represent a bottleneck in the attention mechanism.
> > > >
> > > > *Table 1.* Top-$k$ Latency
> > > >
> > > > | Context | 1024 | 4096 | 8192 | 16384 | 32768 | 65536 |
> > > > | :--- | :---: | :---: | :---: | :---: | :---: | :---: |
> > > > | `torch.topk` | 0.730 | 2.701 | 5.336 | 10.684 | 21.205 | 42.374 |
> > > > | RTop-$k$ | 0.221 | 0.589 | 1.089 | 2.080 | 4.057 | 8.080 |
> > > > | *Ratio of TopK(%)* | *10.51* | *2.10* | *1.96* | *1.90* | *1.03* | *0.51* |
> > > >
> > > > ---
> > > >
> > > > **Q4:**  Limited Applicability and Inherent Distribution Shift: The authors' own experiments in Section 5 demonstrate that applying SFA to pretrained models "introduces a severe distribution shift". The authors' solution is a multi-stage, regularized fine-tuning process, which requires "an additional MSE loss" and even pre-finetuning on another dataset. This implies the method's high adaptation cost.
> > > >
> > > > **A4:** To clarify, the fine-tuning experiments in Section 5 are intended to study how to **retrofit an already dense pretrained model into an SFA-based model**. In this specific setting, we found that a multi-stage procedure with an additional MSE regularization term is helpful.
> > > >
> > > > For practical users, this extra cost is not inherent to SFA. When **the pretrained model itself is trained with SFA from scratch** (as in the pretraining experiments of Section 4.1), downstream fine-tuning proceeds is exactly same with a standard dense model: **a single-stage fine-tuning on the target task, with no additional loss terms, auxiliary datasets, or special procedures required.**
> > > >
> > > > ---
> > > >
> > > > **Q5:** Can the authors provide the corresponding implementation details for the FlashSFA kernel, wrt the above questions in W1?
> > > >
> > > > **A5:** Please see **A1**.
> > > >
> > > > ---
> > > >
> > > > **Q6:** Can the authors provide a revised complexity analysis for the entire SFA block, including the $O(ndk)$ or $O(ndlogd)$ cost of Top-k and the
> > > > $O(n^2d_v)$ cost of the P@V multiplication? How does this revised, complete analysis affect the scaling claims?
> > > >
> > > > **A6:** Please see **A3.**
> > > >
> > > > ---
> > > >
> > > > **Q7:** Given that P@V is a "large proportion" of the FLOPs, why was V not sparsified? Did the authors experiment with this, and if so, what was the impact on performance?
> > > >
> > > > **A7:** Please see **A9**.
> > > >
> > > > ---
> > > >
> > > > **Q8:** The $k^2/d^2$scaling analysis (Eq. 7) relies on an assumption that "supports are balanced across dimensions". Is there empirical evidence for this? What happens to performance and cost if the feature selection is unbalanced?
> > > >
> > > > **A8:** **In practice, we observe that the Top-$k$ coordinate distribution of SFA is close to balanced across layers and heads, so the assumption made in Eq.7 holds well.** In detail we visualize the normalized entropy(where $1.0$ indicates perfectly uniform distribution)of the Top-$k$ index distribution across heads and layers in Qwen3. We found that:
> > > >
> > > > - **$Q$-heads**: For the 16 $Q$-heads, normalized entropy ranges from $0.88$ to $0.98$ (Average: **0.94**).
> > > > - **$K$-heads**: For the 8 $K$-heads, normalized entropy ranges from $0.85$ to $0.97$ (Average: **0.93**).
> > > >
> > > > Overall, these measurements show that SFA maintains nearly balanced feature activations in practice, and the balanced-load approximation in Eq.7 is empirically justified for our setting. We've included this visualization in **Appendix F.**

---

> ### Author Response · Authors · 2025-11-25
>
> **Q9** : Could you clarify how FlashSFA handles these irregular reads from $V$? Specifically, does the overhead of non-contiguous memory access to $V$ diminish the theoretical gains from reduced FLOPs, and how does the kernel implementation mitigate this latency?
>
> We sincerely appreciate your careful re-reading of the paper and your thoughtful comments on the $P @ V$ computation. Your question touches an important implementation detail of FlashSFA.
>
> > Consequently, the accumulation $O_i = \sum_j P_{ij}V_j$ is performed only over the non-zero indices of $P$.
>
> **A9.1:** In both our theoretical analysis and experiments, we have observed that **once $Q$ and $K$ are sparsified, the attention score matrix $P$ also becomes sparse**, so the computation of $P @ V$ becomes a sparse matrix–matrix multiplication (SpMM) with sparse $P$ and dense $V$.
>
> > Could you clarify how FlashSFA handles these irregular reads from $V$?
>
> **A9.2:** **The memory access pattern of $V$, in our implementation, remains largely contiguous** like memory access pattern of $K$.
> - Both $P$ and $V$ are stored row-major in HBM, and the nonzero structure of $P$ is encoded like $Q$.
> - During the $P \cdot V$ SpMM, for each nonzero $P_{ij}$ in $P_{tile}$, the warp cooperatively loads row $j$ of $V$, whose elements are stored contiguously.
> - Binary search is not necessary here since V is dense. As a result, the rows of $V$ are fetched using warp-coalesced, row-vectorized loads guided by the nnz indices in $P$, rather than via scattered, random accesses.
> We have clarified this point in the revised manuscript to make explicit that, although $P$ is sparse, the data layout and traversal order ensure that the reads of $V$ are still effectively contiguous and well-suited to GPU memory hierarchies.
>
> >Specifically, does the overhead of non-contiguous memory access to $V$ diminish the theoretical gains from reduced FLOPs? How does the kernel implementation mitigate this latency?
>
> **A9.3:** Thanks for your insightful suggestion for diagnosing FlashSFA kernel. **We experimentally confirm that memory access to $V$ is negligible and does not diminish the theoretical gains from reduced FLOPs.**
>
> Specifically, we profiled the HBM bandwidth of both the Dense and FlashSFA kernels. Given that **Flash Attention-based kernels are typically compute-bound, arithmetic latency can mask actual memory performance.** Therefore, we conducted **an ablation study by benchmarking a "memory-only" version of the kernels (labeled "w/o compute")**, where the computation logic is disabled to isolate memory throughput. Results are presented in Table 2.
>
> As shown in the "w/o compute" columns, FlashSFA achieves a high physical bandwidth of **919.38 GB/s**. While this is slightly lower than the Dense baseline (**1194.34 GB/s**), it is significantly higher than the effective bandwidth observed in the full kernel (**17.14 GB/s**).
>
> This confirms that the latency of the full kernel is mainly caused by the saturation of compute units, not by memory access to $V$. Consequently, **the primary bottleneck limiting theoretical gains is the algorithmic design of the sparse computation, not memory bandwidth.** We are actively addressing this in our future work by refining the fused kernel with state-of-the-art SpGEMM (for $Q @ K^T$) and SpMM (for $P @ V$) optimizations.
>
> *Table 2.* HBM Bnadwidth Comparison
>
> | **Kernel** | Dense | Dense w/o compute | FlashSFA | FlashSFA w/o compute |
> | :--- | :---: | :---: | :---: | :---: |
> | **HBM Bandwidth(GB/s)** | 14.22 | 1194.34 | 17.14 | 919.38 |
>
> ---
>
> We sincerely thank you for the careful reading and highly constructive suggestions. Your feedback strengthened the completeness and clarity of our paper, and we greatly appreciate the time and expertise you dedicated to improving our work. All corresponding revisions have now been incorporated into the updated version.

---

> > ### Comment · Reviewer_FnGK · 2025-11-25
> >
> > Thank you for the **incredibly thorough and convincing** response. I have read your answers to my questions (as well as the Q&A with other reviewers) and found them very detailed and convincing. I believe my previous questions have been resolved quite well.
> >
> > I am fully satisfied with your explanation regarding the P@V memory access. The update to Algorithm 1 clarifying the contiguous row-reads for $V$ makes the implementation details much clearer. Additionally, the inclusion of strong baselines like MLA and token-sparse methods, along with the entropy analysis in Appendix F, has significantly strengthened the paper's soundness.
> >
> > **I have raised my score to 8.**
> >
> > I have two remaining minor questions for clarification. Please feel free to answer these briefly or conceptually; I do not expect new complex experiments or lengthy write-ups for these, as they are just for better understanding of the method.
> >
> > * **On MLA + SFA Composition**: regarding the new results in Table 1 (Response to Reviewer gwpA), could you briefly clarify where the Top-k sparsification is applied within the MLA architecture? Is it applied to the up-projected (recovered) Key/Value heads, or directly in the compressed latent space?  *My personal curiosity is*: If it's the latter, does enforcing Top-k in a compressed latent space (which is already a dense compression of information) risk conflicting with the MLA's design?
> >
> > * **On Memory Alignment**: Your explanation regarding the coalesced row-reads for $V$ is very convincing. In your current kernel implementation, does this efficiency rely on strict constraints for the head dimension $d_v$ (e.g., must be a multiple of 32 or 128 bytes to align with cache lines)? I am just curious if the performance is sensitive to non-standard head dimensions.
> >
> > I hope this paper will be accepted at this year's ICLR.

---

> ### Author Response · Authors · 2025-11-28
>
> We sincerely appreciate your update and we are glad that hear that we addressed your previous concerns on FlashSFA details. We are happy to further address your remaining questions.
>
> ---
>
> **Q1:** Could you briefly clarify where the Top-k sparsification is applied within the MLA architecture?
> In MLA, the attention score computation during decoding is optimized as:
>
> $$
> \text{Score} = Q \cdot ({W^{UK}}^T \cdot {c^{KV}}^T)=( Q \cdot {W^{UK}}^T ) \cdot {c^{KV}}^T
> $$
>
> Therefore, one of the core parts of MLA is to directly use the latent vector for attention calculation. So we applied SFA to the latent vector $c^{KV} \in \mathbb{R}^{N \times d_c}$ .
>
> ---
>
> **Q2:** Does enforcing Top-k in a compressed latent space (which is already a dense compression of information) risk conflicting with the MLA's design?
>
> We thank the reviewer for this insightful question.
>
> Yes, directly applying Top-$k$ in a highly compressed low-rank space would indeed conflict with MLA's design. However, **we resolve this conflict by increasing the dimension of the latent space $d_c$ to minimize information loss.** Below is our detailed analysis:
>
> We agree that in standard MLA, the latent vector $c_{KV}$ is a dense, extremely compressed representation. According to the theory of **Superposition** [1], neurons in such a constrained space are "polysemantic", distinct features interfere with each other and are superimposed into the same dimensions.
>
> Inspired by recent work on Sparse Autoencoders(SAE)[2,3], we expand the latent dimensionality. The expanded space enables monosemantic representation of features, thereby reducing interference between them.
>
> Ref:
>
> [1] Elhage, Nelson, et al. "Toy models of superposition." arXiv preprint arXiv:2209.10652 (2022).
>
> [2] Gao, Leo, et al. "Scaling and evaluating sparse autoencoders." The Thirteenth International Conference on Learning Representations.
>
> [3] Wen, Tiansheng, et al. "Beyond Matryoshka: Revisiting Sparse Coding for Adaptive Representation." Forty-second International Conference on Machine Learning.
>
> ---
>
> **Q3:** If the performance is sensitive to non-standard head dimensions.
>
> **A3:** Our current kernel **does not rely on strict constraints on the head dimension $d_v$.**
>
> - Each warp cooperatively loads a row $V_j$ **along the feature dimension**, and threads in the warp access consecutive elements of that row.
> - We use vectorized loads whenever possible, and if $d_v$ is not divisible by the vector width, a small “tail” loop handles the remaining elements(1~3) with scalar loads.
> - This tail only affects the last few memory transactions of a row, so the loss in memory coalescing is small compared to the total traffic per row.
>
> In addition, **while “non-standard” head dimensions $d_v$ do not become bottleneck in our implementation**, we do follow a recommended access pattern to better utilize memory bandwidth. Specifically, for `fp16` value matrices $V$, we typically let each thread in a warp load a `half2` vector along the feature dimension. In this way, the warp issues fully coalesced global memory accesses.
>
> ---
>
> Hope this elaboration address your questions and please let us know if there is more to clarify!

---

> > ### Comment · Reviewer_FnGK · 2025-11-28
> >
> > Thank you for your detailed answers. I have no further questions. I will read the related work the author referred to to better understand their responses. This is an **excellent work**, and the amount of work the author demonstrated during the rebuttal and the detailed answers to the questions were impressive.

---

### Official Review · Reviewer_Q1QB · 2025-10-31

**Soundness:** 2
**Presentation:** 2
**Contribution:** 2
**Rating:** 4
**Confidence:** 4

**Summary:**

The authors propose a feature sparse attention to improve the prefill time efficiency. They provide a FlashSFA kernel which integrates their idea into flash attention kernel. They show that models can be trained from scratch to get reasonably close perplexity and downstream accuracy as compared to dense models.

**Strengths:**

1. Feature sparsity is not well explored in literature to the best of my knowledge. This paper does that
2. Opensource FlashSFA kernel that be used by the community.

**Weaknesses:**

1. Lack of baselines.
2. experiments are not convincing. There is degradation in quality when compared to baseline. Requires separate training for retrieval task.

**Questions:**

1. Does FlashSFA also provide improvement in time to next token (TTNT)? From what i understand it can only provide improvement in time to first token(TTFT). In table 2 the speedups refer to TTFT or TTNT?

2.Related to above, given a $k$, what is the memory footprint of storing $k$ indices for one row in CSR format isnt it $3k$? So if we had d=64, $k < 64/3$ is required to get any memory gains.  is that correct understanding?

3. While the paper dismisses sparsity at sequence level as degrading accuracy, we do not see any comparison with such methods in experiments. sparsity in training (NSA / DSA) would be good baselines in my understanding. Can Authors elaborate more on the what are good baslines for FlashSFA in sparse attention techniques.

4. Related to above, the FlashSFA also shows degradation across the board compared to baseline. So it is reasonable to compare against other sparse attention techniques at training time.

5. The fact that we had to train for retreival tasks indepdnently from scratch is a bit worrying. It might have something to do with the information bottleneck in SFA. In principle we are looking for a model that can do all of the tasks. training for specific task is against the foundational model paradigm.

6. How does top-k coordinate distribution look like in practice. Equation 7 assumes that it is balanced. But in practice (atleast in the case of dense models), some channels can show outlier distribution (See AWQ quantization).

7. Also, authors mention that the benefits are multiplicative and their technique can be combined with sequence-sparse attention methods.  Is there some evidence to this effect in authors' experiments / literature.

---

> ### Author Response · Authors · 2025-11-25
>
> We sincerely thank you for your thorough reading and thoughtful comments. We truly appreciate the time and care you invested in reviewing our work. Following your insightful suggestions, we have substantially expanded the discussions and conducted additional experiments on SFA. We deeply value your feedback, and we respectfully address each of your concerns point-by-point below. We hope our clarifications and revisions successfully resolve all the issues you raised and help strengthen the manuscript.
>
> ---
>
> **Q1**: Does FlashSFA also provide improvement in time to next token (TTNT)? From what i understand it can only provide improvement in time to first token(TTFT). In table 2 the speedups refer to TTFT or TTNT?
>
> **A1**: **FlashSFA improves both TTFT and TTNT** as it replaces dense attention with sparse feature attention, accelerating attention computation across both the prefilling and decoding stages.
>
> As shown in **Figure 6(b)** and **Table 1** below, with a relatively small $K$, **FlashSFA does achieves acceleration in the next token prediction stage.** Since SFA **sparsifies the feature vectors themselves**, both **prefilling(TTFT) and decoding(TTNT) are dominated by the same inner-product operations**, the resulting speedups naturally carry over to both stages.
>
> *Table 1.*: Time to next token (ms/step)
>
> | Context           |  4096  |  8192  | 16384  | 32768  | 65536  | KV-cache |
> | :---------------- | :----: | :----: | :----: | :----: | :----: | :------: |
> | **Dense (d=128)** | 0.1452 | 0.2513 | 0.4117 | 0.7577 | 1.4455 |    1     |
> | SFA ($k=16$)        | 0.1348 | 0.217  | 0.3772 | 0.6645 | 1.2963 |  0.688   |
> | SFA ($k=8$)         | 0.1155 | 0.2008 | 0.3269 | 0.5901 | 1.0675 |  0.594   |
>
> ---
>
> **Q2:** Related to above, given a $k$, what is the memory footprint of storing $k$ indices for one row in CSR format isnt it $3k$ ? So if we had d=64, $k<64/3$ is required to get any memory gains. is that correct understanding?
> >Given a $k$, what is the memory footprint of storing $k$ indices for one row in CSR format isnt it $3k$ ?
>
> **A2.1:** Thanks for your question. No, the memory footprint is **not** $3k$ per row. In CSR, a row with $k$ non-zeros stores $k$ values, $k$ column indices, and only **$1$** row-pointer entry.
>
> >So if we had d=64, $k<64/3$ is required to get any memory gains. is that correct understanding?
>
> **A2.2:** In our implementation, **memory gain can be achieved when $k < \frac{2}{3}d$**. The memory savings of SFA compared to the dense model depend on the data precision used to store the `col_indices` array and `row_pointer` array in the CSR matrix.
>
> For a CSR matrix with shape $(N, d)$ where each row has a fixed number of $k$ non-zero values, the required bytes for each component are calculated as follows.
>
> - **`value` array memory**:
>
> $$
> Mem_{value} = (N \times k) \times S_{val}
> $$
>
> - **`indices` array memory**:
>
> $$
> Mem_{indices} = (N \times k) \times S_{idx}
> $$
>
> - **`indptr` array memory** (length is $N+1$):
>
> $$
> Mem_{indptr} = (N + 1) \times S_{ptr}
> $$
>
> Where $S$ denotes the number of bytes for the data format.
>
> Therefore, the total memory consumption of the CSR format is the sum of these three parts:
> $$
> Mem_{csr} = Mem_{value} + Mem_{indices} + Mem_{indptr}
> $$
>
> Substituting the above formulas, we obtain the final memory consumption formula:
>
> $$
> Mem_{csr} = (N \times k \times S_{val}) + (N \times k \times S_{idx}) + ((N + 1) \times S_{ptr})
> $$
>
> This can be further simplified to:
>
> $$
> Mem_{csr} = N \times k \times (S_{val} + S_{idx}) + (N + 1) \times S_{ptr}
> $$
>
> Consequently, compared to a dense matrix of the same shape, the ratio of memory consumption is:
> $$
> Ratio = \frac{Mem_{dense}}{Mem_{csr}} = \frac{N \times d \times S_{val}}{N \times k \times (S_{val} + S_{idx}) + (N + 1) \times S_{ptr}}\approx \frac{d \times S_{val}}{k \times (S_{val} + S_{idx}) + S_{ptr}}
> $$
> As the QK feature dimension in Transformers is generally small, `indices` are typically stored in `int8` format and `indptr` in `int32` format. When we use `fp16/bf16` to store the `value` array:
> $$
> Ratio = \frac{Mem_{dense}}{Mem_{csr}} \approx \frac{d \times 2}{k \times (2 + 1) + 4} = \frac{2d}{3k + 4} \approx \frac{2d}{3k}
> $$
>
> **For the clarity of the paper, we have added the above explanations to Appendix J of the revision.**

---

> > ### Author Response · Authors · 2025-11-25
> >
> > **Q3**: While the paper dismisses sparsity at sequence level as degrading accuracy, we do not see any comparison with such methods in experiments. sparsity in training (NSA / DSA) would be good baselines in my understanding. Can Authors elaborate more on the what are good baslines for FlashSFA in sparse attention techniques.
> >
> > >we do not see any comparison with such methods in experiments. sparsity in training (NSA / DSA) would be good baselines in my understanding.
> >
> > **A3.1:** Following your suggestion, we have added experiments with Longformer (representing fixed token sparsity) and NSA (representing dynamic token sparsity). The results are presented in **Table 2** and **Table 3** below. To demonstrate the orthogonality between SFA and token-level sparsity methods, we also conducted experiments **combining token-level sparsity with SFA**. **Experimental results show that SFA can be combined with Sparsity in Training methods like NSA to further improve efficiency.**
> >
> > *Table 2.* GPT-2 124M Results
> >
> > | Variant | Latency (TTNT)(ms) | Latency (TTFT) | PPL (OWT) | PiQA | LAMBDA | ARC-e | ARC-c | HellaS | Avg |
> > | :--- | :---: | :---: | :---: | :---: | :---: | :---: | :---: | :---: | :---: |
> > | Dense (full) | 17.08 | 16.86 | 17.29 | 42.74 | 22.78 | 28.35 | 8.12 | 19.61 | 24.32 |
> > | Longformer | 6.75 | 7.93 | 18.73 | 41.28 | 21.27 | 28.02 | 7.01 | 18.92 | 23.30 |
> > | Longformer+SFA| **5.23** | **6.18** | 19.30 | 40.75 | 20.54 | 26.39 | 6.63 | 17.24 | 22.31 |
> > | SFA| 14.12 | 9.41 | 18.17 | 41.62 | 21.03 | 28.41 | 7.39 | 19.26 | 23.54 |
> >
> > *Table 3.* Qwen3 0.6B Results
> >
> > | Variant | Latency (TTNT)(ms) | Latency (TTFT) | PPL (Pile) | PiQA | LAMBDA | ARC-e | ARC-c | HellaS | Avg |
> > | :--- | :---: | :---: | :---: | :---: | :---: | :---: | :---: | :---: | :---: |
> > | Dense (full) | 80.84 | 77.65 | 4.66 | 62.47 | 34.82 | 45.41 | 20.35 | 33.95 | 39.40 |
> > | NSA | 9.73 | 20.32 | 4.57 | 62.69 | 35.01 | 45.10 | 20.47 | 34.42 | 39.54 |
> > | NSA+SFA | **8.85** | **17.17** | 4.95 | 60.02 | 33.58 | 42.74 | 18.31 | 32.48 | 37.43 |
> > | SFA | 66.29 | 34.20 | 4.81 | 61.73 | 34.05 | 45.62 | 19.27 | 34.03 | 38.94 |
> >
> > Additional results show that SFA is orthogonal to token sparse method. We have incorperated these results in **Appendix E**. We will merge these experimental results into the final version.

---

> > > ### Author Response · Authors · 2025-11-25
> > >
> > > **Q3**: While the paper dismisses sparsity at sequence level as degrading accuracy, we do not see any comparison with such methods in experiments. sparsity in training (NSA / DSA) would be good baselines in my understanding. Can Authors elaborate more on the what are good baslines for FlashSFA in sparse attention techniques.
> > >
> > > > Can Authors elaborate more on the what are good baslines for FlashSFA in sparse attention techniques.
> > >
> > > **A3.2:** We think that **feature-level** compression methods are more suitable as baselines for comparison, such as low-rank[1] and learned projection methods that compress the dimension of feature vectors. **Table4** bellow shows that under a similar acceleration ratio, **SFA achieves better performance compared to low-rank method.**
> > >
> > > - **Compare with token-level sparsity**: For given $Q/K \in \mathbb{R}^{N \times d}$, token-Level methods (e.g., Longformer, NSA) *reduce the number of tokens ($N$)* involved in the attention computation. While relevant for context length scaling, they are orthogonal to SFA. **SFA reduces the features (non-zeros in $d$)** involved in the inner product, effectively sparsifying the attention mechanism along a different axis.
> > > - **Compare with feature-level baselines**: We also consider methods that reduce the feature dimension ($d$) to be highly relevant baselines, such as Low-Rank Approximation [1] and Learned Projections. These methods compress $Q/K$ vectors from $d$ to a smaller dimension $r$.
> > >
> > > *Table 4.* Benchmark Results to compression baselines under GPT-124M and Qwen3-0.6B
> > >
> > > | Model | Variant | Latency Decode $\downarrow$ (ms) | Latency Forward $\downarrow$ (s) | PPL $\downarrow$ (OWT/Pile) | Acc PiQA $\uparrow$ | Acc LAMBDA $\uparrow$ | Acc ARC-e $\uparrow$ | Acc ARC-c $\uparrow$ | Acc HellaS $\uparrow$ | Acc Avg $\uparrow$ |
> > > | :--- | :--- | :---: | :---: | :---: | :---: | :---: | :---: | :---: | :---: | :---: |
> > > | **GPT2 124M** | Dense (full) | 17.08 | 16.86 | 17.29 | 42.74 | 22.78 | 28.35 | 8.12 | 19.61 | 24.32 |
> > > | | Learned projections ($d=32$) | 8.37 | 7.86 | 20.70 | 39.27 | 19.39 | 25.72 | 6.52 | 14.26 | 21.03 |
> > > | | Low-Rank | 8.93 | 7.99 | 19.89 | 39.81 | 20.04 | 26.47 | 6.89 | 14.99 | 21.64 |
> > > | | SFA ($k=8$) | 14.12 | 9.41 | **18.17** | **41.62** | **21.03** | **28.41** | **7.39** | **19.26** | **23.54** |
> > > | **Qwen3 0.6B** | Dense (full) | 80.84 | 77.65 | 4.66 | 62.47 | 34.82 | 45.41 | 20.35 | 33.95 | 39.40 |
> > > | | Learned projections ($d=64$) | 38.68 | 30.84 | 6.03 | 58.43 | 31.27 | 41.58 | 15.83 | 28.29 | 35.08 |
> > > | | Low-Rank | 40.58 | 32.46 | 5.50 | 59.19 | 31.49 | 41.77 | 15.8 | 30.65 | 35.78 |
> > > | | SFA ($k=16$) | 66.29 | 34.20 | **4.81** | **61.73** | **34.05** | **45.62** | **19.27** | **34.03** | **38.94** |
> > >
> > > As shown in **Table 4**, Low-Rank suffer from significant quality loss due to the aggressive compression of semantic information. In detail:
> > > - GPT-2 124M: SFA achieves a lower Perplexity (**18.17**) compared to Low-Rank (19.89) and Learned Projections (20.70).
> > > - Qwen3 0.6B: SFA maintains a high average accuracy of **38.94**, whereas Low-Rank drops to 35.78 and Learned Projections to 35.08.
> > >
> > > This confirms that SFA's selective sparsity preserves representational capacity much better than the global compression used in Low-Rank baselines, making it the better choice for feature-level acceleration. We've inculded this experiment in **Appendix E**. We will merge these experimental results into the final version.
> > >
> > > Ref: [1] Saxena, Utkarsh, et al. "Eigen Attention: Attention in Low-Rank Space for KV Cache Compression." EMNLP (Findings). 2024.

---

> ### Author Response · Authors · 2025-11-25
>
> **Q4**: Related to above, the FlashSFA also shows degradation across the board compared to baseline. So it is reasonable to compare against other sparse attention techniques at training time.
>
> **A4**: Please see **A3**.
>
> ---
>
> **Q5:** The fact that we had to train for retreival tasks indepdnently from scratch is a bit worrying. It might have something to do with the information bottleneck in SFA. In principle we are looking for a model that can do all of the tasks. training for specific task is against the foundational model paradigm.
>
> **A5:** We wish to clarify that **independent training for retrieval tasks is NOT required for SFA.** Specifically, we conducted new experiments where SFA with Qwen3-0.6B is trained on **general language corpora** rather than specific retrieval tasks (with 4k window) and evaluated it in a **zero-shot** setting on the NIAH task. Results are shown in **Table 5** below.
>
>
> *Table 5*. NIAH accuracy (%) within 4k Context Length.
> | Context Length | 1k | 2k | 3k | 4k | Speedup@4k |
> | :--- | :---: | :---: | :---: | :---: | :---: |
> | Dense (full) | 93 | 87 | 79 | 62 | 1.0x |
> | SFA ($k=8$) | 95 | 90 | 80 | 66 | **1.5x** |
> | SFA ($k=16$) | **96** | **90** | **83** | **71** | 1.2x
>
> As reported in Table 5, SFA not only **works without task-specific supervision but actually outperforms the Dense baseline**:
> *   **Higher Accuracy:** SFA ($k=16$) achieves **71%** accuracy at 4k context, significantly surpassing the Dense model's **62%**. Even with higher sparsity ($k=8$), SFA reaches **66%**.
> *   **Consistent Superiority:** SFA outperforms Dense attention across all context lengths (1k to 4k).
>
> These findings indicate that SFA’s feature sparsification preserves essential information, allowing it to function effectively as a general-purpose foundation model without requiring independent training for retrieval tasks. We've included this experimental results in **Appendix K**.
>
>
> ---
>
> **Q6:** How does top-k coordinate distribution look like in practice. Equation 7 assumes that it is balanced. But in practice (atleast in the case of dense models), some channels can show outlier distribution (See AWQ quantization).
>
> **A6:** As shown in **Figure 7**, we observe that the Top-$k$ coordinate distribution of SFA is **close to be balanced across layers and heads**, so the assumption made in Eq.7 holds well.
>
> Specifically, we calculated the normalized entropy of the Top-$k$ distribution across heads and layers (where a value of $1.0$ represents a perfectly uniform distribution) and results are shown below:
>
> - For the 16 $Q$-heads in Qwen3, the normalized entropy averages $0.94$.
> - For the 8 $K$-heads in Qwen3, the normalized entropy averages $0.93$.
>
> These values are very close to the maximum entropy of $1.0$, indicating that the selected Top-$k$ coordinates are **distributed widely across channels** rather than collapsing onto a few specific outliers. Thus,SFA maintains nearly balanced feature activations in practice, and the balanced-load approximation in Eq.7 is empirically justified for our setting. We've included this discussion in **Appendix F**.
>
> ---
>
> **Q7:** Also, authors mention that the benefits are multiplicative and their technique can be combined with sequence-sparse attention methods. Is there some evidence to this effect in authors' experiments / literature.
>
> **A7:** Please see **A3**.
>
> ---
> We hope that the explanations provided above adequately address your concerns. Following your valuable suggestions, we have revised and improved the manuscript accordingly. We would be grateful to know whether you find the updated version satisfactory, and we remain very willing to clarify any additional questions you may have.

---

> > ### Author Response · Authors · 2025-11-28
> >
> > Dear Reviewer Q1QB,
> >
> > We have carefully prepared a detailed response to address each of your questions. Would you please take a look and let us know whether you find it satisfactory?
> >
> > We note that Reviewer FnGK and gwpA has appreciated our response. We also respectfully suggest that you could re-evaluate our work with the updated explanations and results.
> >
> > Thanks! Have a great day!
> >
> > Authors

---

### Author Response · Authors · 2025-11-25
**Paper Update**

Paper Update

We sincerely thank all reviewers for their detailed reading and valuable comments. We have carefully responded their concerns, and incorporated these suggestions in the updated manuscript.The main revisions are:

- Table 3: update GSM8K results and add new results under *larger model* :**Qwen3-8B**!
- **Appendix F (new!): new analysis of Feature Distribution**
- **Appendix E & G (new!): add extensive new experimental results**
    - Comparison with **Token-Sparse methods** (Longformer, Routing, NSA)
    - Comparison with **Quantization-Aware Training method** (LLM-QAT)
    - Comparison with **KV-Pruning methods** (H2O, Quest, SnapKV)
    - Comparison with **Low-Rank/Kernel methods** (Loki, Performer, Learned Projections), and DeepSeek’s **MLA**.
- Appendix H (new!): add new ablation studies of different sparsity levels ($k$) and feature dimensions ($d$)
- Appendix I (new!): add pre-training curve of SFA on GPT2-124M
- Appendix J (new!): add memory saving theoretical analysis of SFA
- Appendix L (new!): add SVD analysis of Query and Key of pretrained Qwen3 model

---

### Author Response · Authors · 2025-11-28
**Rebuttal Summary**

Dear Program Chairs, Senior Area Chairs, Area Chairs, and Reviewers,

We extend our sincere gratitude to the Program Chairs, Senior Area Chairs, and especially the Area Chairs for their time and effort in coordinating this review process. We also deeply thank all Reviewers for their constructive feedback and active engagement during the discussion phase, which has significantly helped us improve the quality of our work.

To assist the Area Chair in tracking the progress of our rebuttal and the resulting improvements to the manuscript, **we provide a consolidated summary of the discussion phase below.** This summary outlines the key concerns raised, the extensive additional experiments conducted, and the clarifications provided in the revised version. The main revisions are summarized as follows:

**Extensive experiments & analysis.** The primary constructive feedback across reviews centered on the scope of baselines, scalability, and deeper analysis of our methods. In response, we have conducted a comprehensive suite of additional experiments and theoretical analyses to robustly address these points.

1. **Expanded Baselines & Scalability**:
- **Comparison and Integration with Token-Sparse methods**, including Longformer and Native Sparse Attention (NSA), now presented in Appendix E (Raised by Reviewer Q1QB, 5rhv).
- **Comparison and Integration with DeepSeek’s MLA**, now presented in Appendix E (Raised by Reviewer gwpA, 5rhv).
- **Comparison vs. Low-Rank training/compression methods**, including short-embedding and Learned Projections, now presented in Appendix G (Raised by Reviewer Q1QB, 5rhv).
- **Comparison and Integration with KV-Pruning and Kernel methods**, including H2O, Quest, SnapKV, Loki, and Performer, now presented in Appendix G (Raised by Reviewer 5rhv).
- **Scalability Verification (8B Scale)**, We validated SFA on the medium-scale *Qwen3-8B*. Results are now presented in Table 3 (Raised by Reviewer gwpA).

2. **Deeper Analysis & Diagnostics:**
*   **Zero-Shot Long-Context Eval:** We added zero-shot evaluation on long-context tasks (NIAH) using models trained on general corpora, now presented in Appendix K (Raised by Reviewer Q1QB).
*   **Feature Activation Balance**: We conduct a detailed load balance analysis of feature activation, now presented in Appendix F (Raised by Reviewer Q1QB, FnGK).
*   **Memory & Kernel Profiling**: We expanded our system-level analysis to include:
    *   **Bandwidth Profiling:** We provided detailed profiling (w/ and w/o compute) to substantiate our bandwidth savings claims.(Appendix C.4; Raised by Reviewer FnGK).
    *   **Access Pattern Analysis:** Clarification of memory access patterns to address "irregular read" concerns (Appendix C.4; Raised by Reviewer FnGK).
    *   **Kernel Complexity:** A theoretical complexity analysis of the RTop-$k$ kernel (Appendix C.5; Raised by Reviewer FnGK).
    *   **Theoretical memory analysis of CSR Format:** Theoretical memory cost analysis of the Compressed Sparse Row format (Appendix J; Raised by Reviewer Q1QB).
*   **Sensitivity Ablations:** Ablation studies on sparsity levels ($k$) and feature dimensions ($d$), now presented in Appendix H (Raised by Reviewer 5rhv).
*   **Training Stability:** We added pre-training loss curves using the Straight-Through Estimator, now presented in Appendix I (Raised by Reviewer 5rhv).

**Additional clarifications.** Besides experiments, we also provided additional discussions and clarifications to address the concerns of reviewers, including:

*   Clarification on Time-to-Next-Token (TTNT) speedup, now presented in Appendix B.1 (Raised by Reviewer Q1QB).
*   Conceptual explanation of Top-$k$ semantic preservation, now presented in Appendix L (Raised by Reviewer gwpA, 5rhv).
*   Kernel parallelism model, now presented in Appendix C.2 (Raised by Reviewer FnGK).
*   Core data structures details, now presented in Appendix C.3 (Raised by Reviewer FnGK).

---

> ### Author Response · Authors · 2025-12-02
>
> **Writing improvements.** We also refined the presentation in the revised version according to reviewers’ suggestions:
>
> *   Typo correction regarding the dimension size ($d=64$), now corrected in Table 2 (Raised by Reviewer gwpA).
> *   Expanded Appendix sections (E, F, G, H, I, J, K, L) to accommodate the new experimental results and theoretical discussions.
> *   Clarified definitions of "Best results" in tables to distinguish between reference values and competitive baselines (Raised by Reviewer gwpA).
>
> **Outcome & Score.** Several reviewers acknowledged the additional results and clarifications, and **increased their ratings before Nov 27:**
> *   **Nov 25**: Reviewer FnGK increased the score to **8**, noting "this is an **excellent work**".
> *   **Nov 25**: Reviewer gwpA increased the score to **8**, noting "I'm more confident that this is a good paper that should be accepted".
> - Reviewer 5rhv “appreciated all the empirical analysis” and requested further clarification on the rationale for sparse features. We supplied additional evidence from several angles; however, because this was submitted shortly before the review freeze, the reviewer did not have time to reply.
> *   Reviewer Q1QB has not engaged in the discussion. We believe that our new experiments (particularly the zero-shot retrieval in Appendix K and load balance analysis in Appendix F) directly address their concerns.
>
>
> Overall, we believe that the extensive new baselines, scalability verification on 8B models, and deeper system-level analyses have addressed the concerns raised by the reviewers.
>
> We sincerely thank all reviewers for helping us shape this work into a stronger contribution, and we thank you again for your service to the community.
>
> Sincerely,
> Authors

---

### Meta-Review · Area_Chair_TbuM · 2026-01-02

**Summary:**

In the first review rounds, the reviewers acknowledged the strengths of the paper being:
1. contributing to open-source community
2. solid theoretical and experimental work
3. simplicity of the work

The following weaknesses of the work were also raised by the reviewers:
1. Missing baseline and convincing experiments, as some metrics were not improving.
2. Analysis is missing several details.
3. Lack of comparison with MLA and other methods.

Most of the weaknesses were already addressed in the rebuttal.

**Reviewer Concerns:**

In the rebuttal, the author clearly addressed
1. Baselines such as NSA + SFA and Longformer + SFA.
2. Promise to release the kernel upon acceptance.
3. Missing analysis mentioned by reviewer FnGK.
4. Adding experiments on models other than Qwen and comparison with MLA.

Overall, most of the weaknesses, including the questions, were well addressed in the rebuttal.

**Reviewer Scores:**

Reviewers FnGK and gwpA will increase the score. Reviewer 5rhv might increase the score as the follow-up comments were addressed.

---

### Decision · Program_Chairs · 2026-01-26

Accept (Poster)